# A FUNCTION CENTRIC PERSPECTIVE ON FLAT AND SHARP MINIMA

## ABSTRACT

Flat minima are widely believed to correlate with improved generalisation in deep neural networks. However, this connection has proven more nuanced in recent studies, with both theoretical counterexamples and empirical exceptions emerging in the literature. In this paper, we revisit the role of sharpness in model performance, proposing that sharpness is better understood as a function-dependent property rather than a reliable indicator of poor generalisation. We conduct extensive empirical studies, from single-objective optimisation to modern image classification tasks, showing that sharper minima often emerge when models are regularised (e.g., via SAM, weight decay, or data augmentation), and that these sharp minima can coincide with better generalisation, calibration, robustness, and functional consistency. Across a range of models and datasets, we find that baselines without regularisation tend to converge to flatter minima yet often perform worse across all safety metrics. Our findings demonstrate that function complexity, rather than flatness alone, governs the geometry of solutions, and that sharper minima can reflect more appropriate inductive biases (especially under regularisation), calling for a function-centric reappraisal of loss landscape geometry.

## 1 INTRODUCTION

Neural network architectures with different implicit biases are known to exhibit distinct geometric properties around the loss landscape minima, with flatness often associated with improved generalisation performance via reduced generalisation gaps (Li et al., 2018). This desirability has been linked to the idea that flat minima correspond to wide error margins and thus increased robustness – in line with Occam's Razor (Hochreiter & Schmidhuber, 1994). Empirical and theoretical studies have sought to support this perspective (Kaddour et al., 2022; Foret et al., 2021; Petzka et al., 2021), reinforcing the view that flatter solutions lead to better generalisation. However, the benefits of flat minima have also been questioned. Dinh et al. (2017) showed that flat minima, under commonly used definitions and metrics, can be arbitrarily sharpened via reparameterisation, without changing the model's function or generalisation properties. This motivated the development of reparameterisation-invariant sharpness metrics, such as the Fisher-Rao-Norm (Liang et al., 2019) and Relative-Flatness (Petzka et al., 2021) which reaffirmed the correlation between flatness and generalisation.

Flatness has also been associated with benefits such as improved representation transfer (Liu et al., 2023) and the effects of architectural choices such as residual connections (Li et al., 2018). Notably, optimisation methods such as Sharpness Aware Minimization (SAM) (Foret et al., 2021), which improve generalisation in the vision domain, explicitly aim to bias training toward flatter minima. Yet generalisation is only one dimension of model quality. Safety-critical properties, such as robustness to average-case perturbations (Hendrycks & Dietterich, 2019), calibration (Guo et al., 2017) and functional diversity (Wang et al., 2024), are essential for reliable deployment. However, their relationship to flatness remains underexplored. In particular, it is unclear whether flatter solutions consistently support better safety, or whether high-performing models on these dimensions may instead occupy sharper regions of the loss landscape.

In this paper, we investigate this question through a function-centric lens: we hypothesise that the geometry of a solution reflects the complexity of the learned function, rather than directly determining performance. From this perspective, sharper minima may not indicate overfitting, but instead reflect more expressive or better-regularised solutions, particularly in high-dimensional learning tasks.

We begin with seven standard single-objective optimisation problems, where global minima are known and can be geometrically compared. These reveal that optimal solutions can be either sharp or flat, depending on the intrinsic complexity of the objective: some functions (e.g., Sphere) have flat global minima, while others (e.g., Rosenbrock) have inherently sharp global mimima. This indicates that the geometry of the solution space is tied to function complexity, not optimality.

We then scale our analysis to high-dimensional problems, and use the CIFAR (Krizhevsky & Hinton, 2009) and TinyImageNet (Le & Yang, 2015) datasets to train the ResNet (He et al., 2016) VGG (Simonyan & Zisserman, 2015), and ViT (Dosovitskiy et al., 2021) architectures. We compare baseline models to those trained with standard regularisation techniques (SAM, weight decay, and data augmentation), evaluating each using reparameterisation-invariant sharpness metrics, generalisation performance, and safety-critical evaluations: expected calibration error, average-case perturbation robustness, and functional agreement.

Our findings provide strong empirical support for a function-centric view of sharpness: models trained with regularisation typically converge to sharper minima, and often outperform their flatter, unregularised counterparts across safety and generalisation metrics (Figure 1). This indicates that regularisation increases the complexity of the learned function, leading to sharper but more effective solutions. However, it is important to note that distinct trends emerge in each control condition, some with a preference for flatter solutions and others not suggesting the emergence of the Simpson's Paradox (Simpson, 1951), which means that minima geometry requires a nuanced view over a one-size-fits-all preference for flatness. While SAM and related methods were originally motivated by the goal of encouraging flatness, we show that their benefits frequently arise despite increasing sharpness. Together, these results challenge the assumption that flatness is inherently beneficial and support a reappraisal of sharpness through the lens of function complexity.

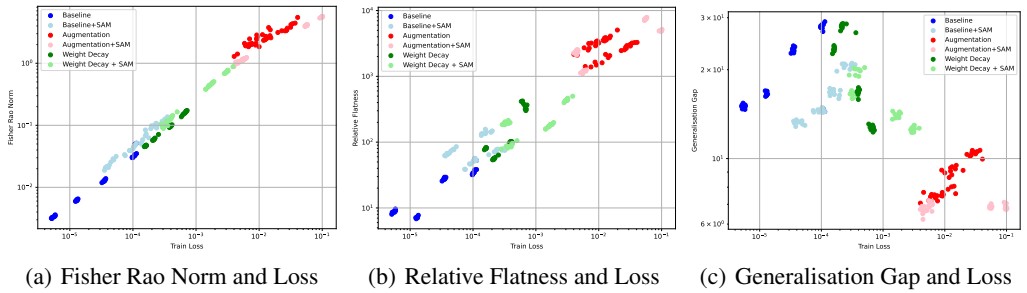

| (a) Fisher Rao Norm and Loss | (b) Relative Flatness and Loss | (c) Generalisation Gap and Loss |

Figure 1: Scatter plots of 240 converged minima for ResNet-18 on CIFAR-10 across batch size 128, 256 and learning rate $10^{-3}, 10^{-2}$: (a) Fisher–Rao norm vs. train loss, (b) Relative Flatness vs. train loss, and (c) generalisation gap vs. train loss (log scale). Full results in Appendix E.2.

Concretely, we make the following contributions:

- We advance a function-centric interpretation of sharpness, where the geometry of minima reflects the complexity of the learned function rather than serving as a universal proxy for generalisation.

- We provide empirical evidence from both toy optimisation problems and high-dimensional deep learning tasks that sharper minima can coincide with better generalisation, calibration, and robustness, particularly under regularisation.

- We show that widely used regularisation techniques (e.g., SAM, weight decay, augmentation) often induce sharper minima, contradicting the assumption that regularisation generally promotes flatter solutions.

- We demonstrate that sharpness cannot be meaningfully compared across architectures or tasks without accounting for function complexity and implicit bias, cautioning against overgeneralised geometric claims.

Our findings can be summarised as follows:

1. Sharpness varies across global minima in single-objective optimisation, reflecting function complexity rather than solution quality. Regularised models, on high dimesnional problems typically converge to sharper minima, yet often achieve better generalisation, calibration, robustness, and functional consistency than flatter unregularised baselines.

2. We reconcile SAM's local robustness objective with increased global sharpness, aligning with a function-centric view of geometry.

3. Our results support a function-centric view of sharpness: solution geometry is shaped by the complexity of the learned function and the model's inductive biases. Crucially there exists no clear goldilocks zone for sharpness across architectures and datasets as sharpness is dependant on function complexity and implicit bias.

## 2 RELATED WORK

Hochreiter & Schmidhuber (1997) presented seminal empirical evidence that neural networks adhered to Occam's Razor. They showed that a flat minimum search algorithm using a second-order hessian approximation could yield the smallest generalisation gap on two-class classification problems. Therefore, due to the observed empirical relation of flatness and generalisation it was thought that the antipodal sharp minima were undesirable. The importance of flatness in more complex learning tasks was later reaffirmed by Li et al. (2018) who introduced landscape visualisation to study the geometry of deep networks. They argued that skip connections prevent explosions of non-convexity, helping to avoid chaotic plateaus often associated with sharp minima. Building on this, Sharpness Aware Minimisation (Foret et al., 2021) was proposed as an optimisation method (motivated by Hochreiter & Schmidhuber (1997)) that explicitly aims to reduce sharpness in the loss landscape. SAM has yielded strong empirical performance gains over traditional optimisation (Foret et al., 2021). However, some literature has challenged this interpretation, arguing that SAM does not necessarily find flatter minima (Wen et al., 2023). The necessity of flatness for generalisation has also been questioned more fundamentally. Notably, Dinh et al. (2017) demonstrate that sharpness can be arbitrarily increased through reparameterisation without affecting generalisation, casting doubt on the intrinsic value of flatness. In response, reparameterisation-invariant sharpness metrics were developed (Petzka et al., 2021) and have since been used to reaffirm the correlation between flatness and generalisation. Together, these developments highlight a conceptual tension: while sharpness was shown to be manipulable through reparameterisation and thus not an intrinsic property of the learned function, flatness is still widely used as a desirable indicator of generalisation.

In this paper, we revisit the role of flatness in deep learning. We argue that the geometry of a neural network's minimum should reflect its capacity to match the complexity of the function represented by the data, rather than conform to a prior preference for flatness. From this function-centric view, regularisation improves performance not by flattening the loss landscape, but by enabling the learning of more complex functions – functions that are harder to learn, often require more intricate decision boundaries, and are frequently associated with sharper minima. Contrary to the view that sharpness signals poor generalisation, we show that sharper solutions can emerge precisely when models generalise better. We propose that sharpness reflects task complexity and inductive bias, challenging its conventional role as a proxy for generalisation.

## 3 SHARPNESS, GENERALISATION AND SAFETY CRITICAL EVALUATIONS

**Sharpness Metrics:** We employ three established measures of sharpness from the literature, namely Fisher-Rao Norm (Liang et al., 2019), Relative Flatness (Petzka et al., 2021), and average-case SAM-Sharpness (Foret et al., 2021). Formal definitions are provided in Appendix Section B. Hessian-based metrics, such as the eigenvalue of the Hessian and the trace of the Hessian, were shown not to be reparameterisation invariant as they can be manipulated via linear reparameterisations. These reparameterisations do not change the function of the model but can make the minima sharper Dinh et al. (2017), undermining relationships between generalisation and flat minima. As a result, we focus on two sharpness metrics in particular, Fisher-Rao Norm (Liang et al., 2019) and Relative Flatness (Petzka et al., 2021) that are reparametrisation invariant to ensure that our study, and its findings are robust (to reparametrisations).

**Calibration:** Calibration measures how well a model's predicted confidence aligns with its true likelihood of correctness. Deep networks, including ResNets, have been shown to be systematically overconfident (Guo et al., 2017), reducing trust in their predictions. We measure calibration using Expected Calibration Error (ECE) (Guo et al., 2017), where lower values indicate better calibration and higher trustworthiness.

**Functional Diversity:** Functional diversity reflects how similar neural networks are in their representation space (Wang et al., 2024; Mason-Williams et al., 2024b; Mason-Williams, 2024). Prior work has linked diversity in function space to improved ensemble performance (Fort et al., 2020; Lu et al., 2024), while others argue that representation convergence can also benefit ensembling (Wang et al., 2024). We quantify functional similarity using prediction disagreement on the test set, which captures how often models disagree on their outputs. Lower disagreement implies that models tend to agree more on individual predictions given the same training data, indicating stronger functional similarity. We interpret this agreement as a desirable property, reflecting stability in the learned function and robustness to training stochasticity.

**Robustness:** Robustness assesses how well a model performs under distribution shift or input perturbations, which is crucial for deployment in safety-critical settings (Hendrycks & Dietterich, 2019). We evaluate robustness on CIFAR10-C and CIFAR100-C (Hendrycks & Dietterich, 2019), which include common corruptions such as impulse noise, JPEG compression, and contrast distortions. Performance is quantified via mean corruption accuracy; higher values indicate greater robustness. The perturbations explored represent average-case perturbation over worst-case perturbations that are typically explored in adversarial robustness studies(Hendrycks & Dietterich, 2019)

Each of the evaluation axes above extends beyond accuracy and captures different aspects of model. We argue that these metrics are essential for evaluating models in real-world, safety-critical contexts. Moreover, they provide a broader view of generalisation that complements geometric analyses such as sharpness. We formally define and provide additional details on all evaluation protocols in Appendix C.

## 4 SINGLE-OBJECTIVE OPTIMISATION

We posit that the sharpness reached by a model depends on the geometric properties of the function it is trained to approximate. To illustrate that loss-landscape geometry is tied to solution complexity, we begin with a toy setting: single-objective optimisation. Toy settings have been used to study geometric properties of neural networks such as Huang et al. (2020) which used the Swiss Roll dataset to explore generalisation and flat minima. Consider Himmelblau's function in equation 1 (visualised in Figure 2). It has four global minima whose local geometry differs markedly (Table 1), yet each achieves zero loss. Thus, no minimum is intrinsically preferable from an optimisation objective standpoint. Under flatness-centric views, flatter minima would be deemed superior; however, any network that represents the target function can plausibly converge to any of these minima. Flatness is therefore not a necessary criterion for optimality in this setting.

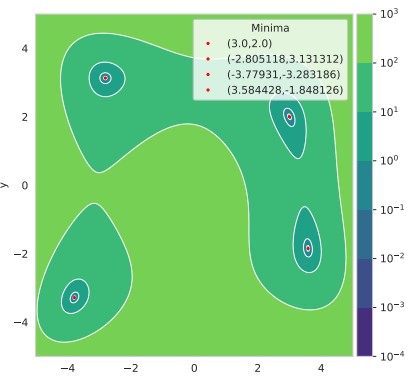

Figure 2: Himmelblau's function landscape with four global minima in red.

$$f(\boldsymbol{x}, \boldsymbol{y}) = (x^2 + y - 11)^2 + (x + y^2 - 7)^2 \tag{1}$$

| Global Minimum | Condition Number | Hessian Trace | Hessian determinant | Max Eigenvalue |
|---|---|---|---|---|
| (3.0, 2.0) | 3.200 | 108.000 | 2116.000 | 82.284 |
| (-2.805118, 3.131312) | 1.242 | 145.39 | 5222.890 | 80.550 |
| (-3.77931, -3.283186) | 1.892 | 204.500 | 9460.560 | 133.786 |
| (3.584428, -1.848126) | 3.674 | 134.110 | 3024.540 | 105.419 |

Table 1: Local geometric properties at the four global minima of Himmelblau's function.

Moving beyond this example, we examine a set of single-objective problems with a single global minimum. Figure 3 visualises six such functions (definitions in Appendix A). Each exhibits a distinct landscape, implying different local curvature at its global minimum. Table 2 reports sharpness statistics at the global minimum. For instance, the Sphere function is the flattest across metrics, whereas functions with more intricate landscapes (e.g., Rosenbrock, Beale, Booth) have sharper optima. Accurately representing these objectives therefore entails reaching minima with geometry commensurate to the function's complexity.

Table 2: Sharpness at the global minimum for six single-objective optimisation functions.

| Function | Condition Number | Hessian Trace | Hessian determinant | Max Eigenvalue |
|---|---|---|---|---|
| Sphere | 1.000 | 4.000 | 4.000 | 2.000 |
| Rosenbrock | 2508.010 | 1002.000 | 400.000 | 1001.600 |
| Rastrigin | 1.000 | 793.568 | 157438.000 | 396.784 |
| Beale | 162.473 | 49.281 | 14.766 | 48.980 |
| Booth | 9.000 | 20.000 | 36.000 | 18.000 |
| Three hump camel | 2.784 | 6.000 | 7.000 | 4.414 |

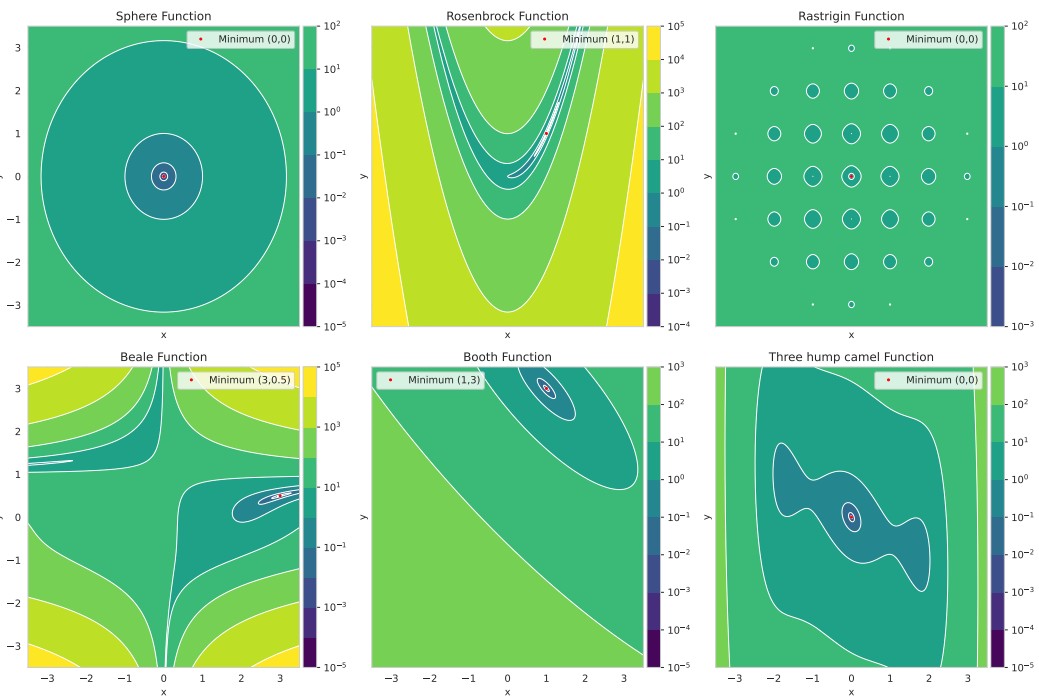

Figure 3: Landscapes for six single-objective functions.

We next fit an MLP to each objective using the same initialisation and average over ten models. As shown in Figure 4, the sharpness of local minima encountered during training reflects the sharpness of the global optimum: with a fixed training budget, model sharpness, training loss, and generalisation gap are governed by the complexity of the target function (cf. Figure 3). Although

absolute generalisation gaps differ across objectives, they exhibit similar relative reductions over training. Appendix A.2 further shows that matching final loss across functions still yields different sharpness levels, as expected from their intrinsic geometry. Consequently, flatness is desirable only when demanded by the target function (e.g., Sphere). Seeking flat solutions for intrinsically sharper objectives (e.g., Rosenbrock) is suboptimal: their complexity is consistent with the need for tighter decision boundaries and thus sharper minima. It is important to note that this section is purely illustrative of how neural network minima geometry can relate to function complexity and that this analysis in the regression case would not hold for sharpness metrics such as Relative Flatness due to their requirement for locally constant labels Petzka et al. (2021). In the following section, we see how our findings in this toy setting extend to reparametrisation invariant metrics, Fisher Rao norm, and Relative Flatness in classification settings where locally constant label conditions hold.

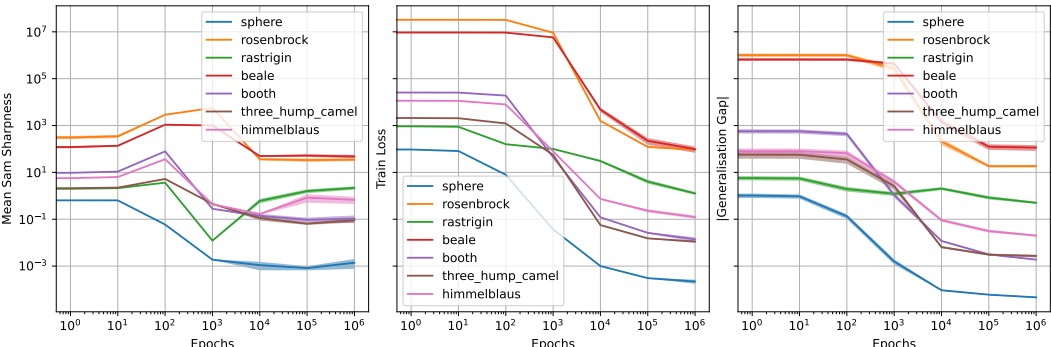

Figure 4: Training an MLP on single-objective problems over epochs: mean sharpness, training loss, and absolute generalisation gap (averaged over 10 runs).

Furthermore, in Appendix Section I we show how arbitrarily increasing the complexity of training data in a classification task results in a model reaching a sharper minima under Fisher Rao norm and Relative Flatness. The findings from this experiment confirm the insights gained on function complexity and geometric properties in this toy setting.

## 5 HIGH-DIMENSIONAL OPTIMISATION PROBLEMS

Building on the view that flatness reflects the complexity of the function being fit, we extend our analysis to high-dimensional settings and ground it in the vision domain. In practice, deep vision neural networks are routinely trained with regularisation (Goodfellow et al., 2016; Kukačka et al., 2017), yet why specific regularisers improve generalisation remains only partially understood despite extensive prior work (Tian & Zhang, 2022; Moradi et al., 2020; Santos & Papa, 2022) – making vision an ideal test-bed to study how geometry relates to reliability (calibration, robustness, and prediction agreement) at scale. Our contribution is to examine these phenomena through the lens of solution (function-space) complexity, explicitly linking geometry to both generalisation and safety-relevant measures. This function-centric perspective offers a complementary reading of flat and sharp minima.

**Function Complexity:** Occam's Razor, or the Principle of parsimony, formally states that of two competing hypotheses, $\mathcal{H}$ and $\mathcal{H}'$ which both adequately describe an event event, $\mathcal{E}$, and are composed of assumptions, $\mathcal{A}$, where the number of assumptions is bounded by $\mathcal{K}$ and $\mathcal{J}$ where $\mathcal{K} < \mathcal{J}$ and $\{\mathcal{A}_1, \mathcal{A}_2, ..., \mathcal{A}_\mathcal{K}\} \in \mathcal{H}$ and $\{\mathcal{A}_1, \mathcal{A}_2, ..., \mathcal{A}_\mathcal{J}\} \in \mathcal{H}'$, there should be a preference towards the hypothesis which has the fewest assumptions, $\mathcal{A}_\mathcal{K}$, (Good, 1977). For neural networks, this has been understood, in relation to the minimum descriptive length (MDL), that neural networks with flatter minima make fewer assumptions and can be described with less precision as they remain approximately constant under perturbation and therefore, in line with Occam's Razor, should be preferred Hochreiter & Schmidhuber (1994). As a result, it can be understood through an extension of Occam's Razor and MDL that minima sharpness represents the quantity of assumptions required, meaning that sharp minima have a longer MDL, and therefore (for better or worse) represent a more complex function. Through this lens, we study how sharpness (function complexity) is governed

by regularisation to better understand generalisation and training dynamics. We further this by connecting this complexity study to safety-relevant evaluations such as robustness, calibration, and prediction consistency.

Despite the work of Dinh et al. (2017), flat minima are still considered important for improved generalisation (Han et al., 2025; Petzka et al., 2021; Lee & Yoon, 2025; Cha et al., 2021; Zhao et al., 2022). However, the connection of minima geometry to safety metric evaluation and function complexity remains underexplored. Existing perspectives in the flatness literature suggest that neural networks with small generalisation gaps - and, by extension, strong safety metric performance - should be found at flatter minima, however, our single-objective analysis indicates a different picture: regularisation may be able to yield sharper minima when the learned functions is represented with more precision. We therefore examine, in a controlled manner, how commonly used regularisers affect sharpness and the corresponding safety evaluations across matched seeds.

More formally, given a training control (regulariser) $c$, we examine how it impacts sharpness, and what are the corresponding safety evaluations. Let the set of controls (training conditions) be $\mathcal{C} = \{\text{Baseline, Baseline+SAM, Aug, Aug+SAM, WD, WD+SAM}\}$. Let $\mathcal{M} = \{\text{FR}, \text{RF}, \text{SAM}\}$ denote sharpness metrics (Fisher–Rao, Relative Flatness, SAM sharpness; lower is flatter), and let $\mathcal{R} = \{\text{Acc}_{\text{clean}}, \text{Acc}_{\text{corr}}, \text{ECE, Disagree}\}$ denote evaluation metrics (test accuracy, corruption-robust accuracy where available, calibration, prediction disagreement). We run seeds $i \in \{0, \ldots, 9\}$ with identical initialisation and data order across controls.

For each control $c \in \mathcal{C}$ and seed $i$, we record $S_{i,m}^{(c)}$ ($m \in \mathcal{M}$), $R_{i,r}^{(c)}$ ($r \in \mathcal{R}$). We report per-control, per-metric summaries as means across seeds:

$$\bar{S}_m^{(c)} = \frac{1}{n} \sum_{i=0}^{n-1} S_{i,m}^{(c)}, \qquad \bar{R}_r^{(c)} = \frac{1}{n} \sum_{i=0}^{n-1} R_{i,r}^{(c)},$$

and present mean $\pm$ SEM Belia et al. (2005) across seeds.

**Hypothesis:** Regularisation tends to increase sharpness (larger $\bar{S}_m^{(c)}$ than Baseline), while the corresponding evaluations often improve (higher accuracy metrics; lower ECE and disagreement).

### 5.1 EXPERIMENTAL SETUP

We adopt the notation above. We run $n = 10$ matched seeds; for each seed, all controls share the same initial weights and data order. This ensures that models trained under different controls start from the same point in the loss landscape and, in principle, could traverse to (and even reach) the same minima, enabling controlled geometric comparisons. Each control is applied independently; all other training details (optimiser, schedule, epochs, etc.) are held fixed across controls. Our objective is to characterise, under controlled conditions, the geometric and safety effects of regularisation controls, not to optimise for state-of-the-art performance. We define the controls as follows.

**Baseline**: Vanilla training without additional regularisation. For each architecture/dataset, the exact baseline configuration is specified in Appendix D. The baseline serves as the reference for geometric and safety metrics against which all regularised controls are compared.

**Weight Decay, Augmentation and SAM:** We consider weight decay ($5 \times 10^{-4}$), data augmentation (random rotation and crop), and SAM, applied individually or in combination as defined in $\mathcal{C}$. We record their effect on sharpness metrics ($\mathcal{M}$) and safety evaluations ($\mathcal{R}$) under the matched-seed setup.

## 6 RESULTS

We present results for ResNet18 trained on CIFAR10, CIFAR100, and TinyImageNet. For each control in $\mathcal{C}$, we report geometric sharpness metrics ($\mathcal{M}$) and reliability-relevant evaluations ($\mathcal{R}$) across 10 matched seeds. Appendix D details training and sharpness metric settings per dataset. Results for VGG and ViT architectures appear in Appendix F and G, confirming the broader trends observed here. Tables 3, 5, 7 below summarise how each training control affects sharpness and safety evaluations. Means $\pm$ SEM are reported per metric. TinyImageNet results exclude Corruption

Accuracy and Relative Flatness due to metric inapplicability. Additional results for batch size (256 and 128) and learning rate ($1e^{-3}$ and $1e^{-2}$) sweeps for ResNet and VGG are in Appendix E.2 and F.1, further confirming the trends observed here. In Appendix Section H we explore how increasing the radius $\rho$ hyperparameter of SAM can increase a model's sharpness over the baseline and in turn improve performance. Finally, in Appendix Section I, we artificially increase the function complexity of training data and observe how minima become sharper when classes have increasingly disjoint examples. Here, we see that function complexity and minima geometry are inherently related.

Table 3: Results for ResNet18 trained on CIFAR10. Bolded values indicate the best performance per metric. For sharpness metrics, lower values correspond to flatter models.

| Condition | Generalisation Gap | Test Accuracy | Test ECE | Corruption Accuracy | Prediction Disagreement | Fisher Rao Norm | SAM Sharpness | Relative Flatness |
|---|---|---|---|---|---|---|---|---|
| Baseline | 28.050 ±0.175 | 0.720 ±0.002 | 0.186 ±0.001 | 58.614 ±0.201 | 0.282 ±0.001 | 0.032 ±0.001 | 1.366E-05 ±1.206$E-06$ | 34.607 ±0.757 |
| Baseline + SAM | 20.588 ±0.125 | 0.794 ±0.001 | 0.108 ±0.001 | 66.342 ±0.164 | 0.168 ±0.000 | 0.107 ±0.006 | 5.823E-05 ±9.056$E-06$ | 75.093 ±1.693 |
| Augmentation | 10.399 ±0.067 | 0.886 ±0.001 | 0.077 ±0.001 | 68.755 ±0.219 | 0.121 ±0.001 | 3.940 ±0.207 | 1.905E-01 ±2.203$E-02$ | 2903.220 ±89.243 |
| Augmentation + SAM | **6.864** ±0.038 | **0.908** ±0.000 | **0.014** ±0.001 | **71.419** ±0.283 | **0.069** ±0.000 | 5.571 ±0.035 | 1.303E-01 ±1.547$E-02$ | 4970.972 ±30.139 |
| Weight Decay | 27.942 ±0.196 | 0.721 ±0.002 | 0.174 ±0.002 | 58.562 ±0.227 | 0.281 ±0.001 | 0.065 ±0.004 | 3.391E-05 ±4.494$E-06$ | 59.767 ±3.009 |
| Weight Decay + SAM | 19.788 ±0.149 | 0.802 ±0.001 | 0.096 ±0.001 | 67.079 ±0.117 | 0.162 ±0.001 | 0.127 ±0.006 | 8.733E-05 ±1.430$E-05$ | 88.807 ±2.336 |

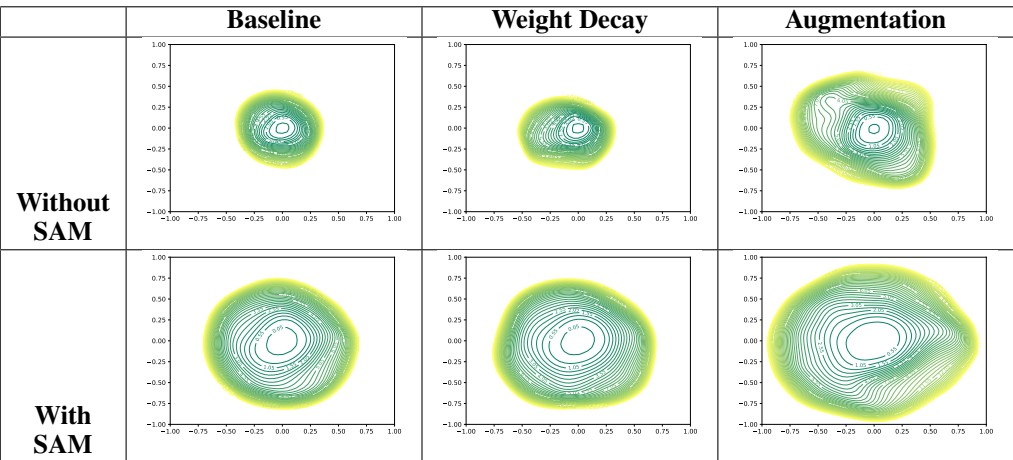

Table 4: Loss landscape visualisation Li et al. (2018) of ResNet18 landscape on CIFAR10 exploring the loss in the domain of the perturbations $[-1, 1]^2$ with 51 steps in both directions.

**Regularisers Increase Sharpness and Improve Evaluations.** Across all CIFAR datasets and architectures, we observe a recurrent trend: the Baseline condition yields the flattest minima (lowest values across FR, RF, SAM), yet performs worst on test accuracy and safety-relevant metrics: calibration (ECE), robustness (Corruption Accuracy), and functional consistency (Prediction Disagreement) (Tables 3, 5). Conversely, controls with stronger regularisation tend to yield sharper solutions while also achieving better evaluations. This challenges the conventional view that flatter minima are inherently preferable, and instead supports the function-centric perspective that sharper minima can reflect more complex, well-generalising solutions. Crucially, we also find that sharper minima can empirically yield better safety-relevant performance.

**Limitations of Loss Landscape Visualisations.** Loss landscape visualisations (Figures 4, 6), produced using the method of Li et al. (2018), qualitatively illustrate that regularisation – especially SAM – alters the geometry of the solution. These plots often appear broader in some directions, even when sharpness metrics increase. This apparent mismatch underscores the limitations of low-dimensional loss surface plots, which capture only 2D projections of high-dimensional landscapes. In contrast, sharpness metrics reflect geometric properties beyond local projections. While visualisations can help convey functional changes, metric-based evaluations provide a more consistent and interpretable picture of sharpness.

**SAM Does Not Always Flatten:** Contrary to prior claims that SAM finds flatter solutions (Foret et al., 2021; Cha et al., 2021), our results show that SAM often increases sharpness across metrics and conditions (Tables 3, 5 and 7, as well as Appendix Sections F and G). Notably, Augmentation+SAM achieves the best performance across evaluations while also being the sharpest model. There are limited exceptions; for example, SAM Sharpness decreases for Aug+SAM on CIFAR10 and CIFAR100 (Tables 3, 5), but these are not consistent across metrics. On more complex datasets (TinyImageNet; Table 7), SAM can sometimes lead to flatter solutions, though this behaviour is again inconsistent. Overall, these findings show that SAM supports the learning of higher-performing functions that may reside in sharper regions of the loss landscape. In Appendix Section H, we show how modifying the $\rho$ radius hyperparameter of SAM can directly modify the sharpness of the minima found at the end of training. In these results, we reaffirm that the best models navigate to even sharper minima than the baseline as the perturbation radius grows.

Table 5: Results for ResNet18 trained on CIFAR100.Bolded values indicate the best performance per metric. For sharpness metrics, lower values correspond to flatter models.

| Condition Condition | Generalisation Gap | Test Accuracy | Test ECE | Corruption Accuracy | Prediction Disagreement | Fisher Rao Norm | SAM Sharpness | Relative Flatness |
|---|---|---|---|---|---|---|---|---|
| Baseline | 47.010 ±0.166 | 0.530 ±0.002 | 0.220 ±0.001 | 38.760 ±0.085 | 0.452 ±0.000 | 0.294 ±0.028 | 2.607E-04 ±3.147$E-05$ | 32.085 ±0.313 |
| Baseline + SAM | 44.421 ±0.168 | 0.556 ±0.002 | 0.191 ±0.002 | 41.888 ±0.098 | 0.410 ±0.000 | 0.399 ±0.014 | 4.231E-04 ±4.973$E-05$ | 123.791 ±4.185 |
| Augmentation | 29.642 ±0.133 | 0.697 ±0.002 | 0.185 ±0.001 | 44.613 ±0.169 | 0.288 ±0.001 | 3.587 ±0.150 | 1.110E-01 ±9.173$E-03$ | 2766.925 ±178.669 |
| Augmentation + SAM | **28.999** ±0.092 | **0.705** ±0.001 | 0.145 ±0.001 | **45.428** ±0.217 | **0.269** ±0.000 | 4.179 ±0.032 | 1.081E-01 ±1.636$E-02$ | 4196.832 ±52.606 |
| Weight Decay | 47.838 ±0.301 | 0.521 ±0.003 | **0.099** ±0.005 | 37.868 ±0.265 | 0.474 ±0.001 | 0.861 ±0.116 | 5.192E-04 ±8.009$E-05$ | 136.969 ±7.484 |
| Weight Decay + SAM | 45.644 ±0.117 | 0.543 ±0.001 | 0.106 ±0.002 | 40.604 ±0.222 | 0.444 ±0.001 | 1.788 ±0.069 | 1.528E-03 ±1.427$E-04$ | 360.271 ±16.190 |

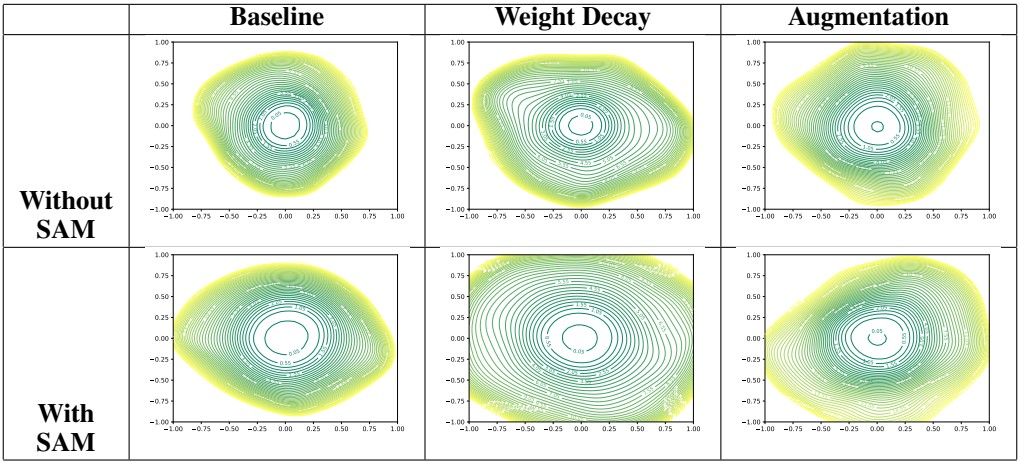

Table 6: Loss landscape visualisation Li et al. (2018) of ResNet18 landscape on CIFAR100 exploring the loss in the domain of the perturbations $[-1, 1]^2$ with 51 steps in both directions.

**Reconciling SAM's Objective with Increased Sharpness.** While SAM is commonly understood as a flatness-promoting method (Foret et al., 2021), its objective encourages local robustness rather than global flatness. Specifically, SAM minimises the loss at the worst-case perturbation within a small neighbourhood around the current weights, thereby promoting low curvature in that vicinity. However, this does not guarantee low values across all global or reparameterisation-invariant sharpness metrics. Our findings – where SAM often increases Fisher–Rao norm, Relative Flatness, and SAM-sharpness – highlight that sharper solutions can still emerge, especially when the model learns more complex or expressive functions. This suggests that SAM enables good generalisation and safety not solely by flattening, but by guiding the model to solutions that are robust in important local directions, even if globally sharp under broader measures. To our knowledge, this is the first work to systematically document that SAM can increase multiple sharpness metrics and to interpret this effect through the lens of local robustness, helping to reconcile SAM's flatness-based motivation with empirically sharper solutions.

**Safety Properties Can Exist at Sharper Minima.** Across all CIFAR datasets, we consistently observe that the Baseline control yields the flattest solutions, yet performs worst on safety-relevant evaluation. In contrast, the controls that achieve the best performance on these metrics are always sharper than the Baseline. These results suggest that sharper minima can coincide with improved safety properties, indicating that sharpness may in fact be an important factor in achieving reliable models. One possible explanation is that sharper minima correspond to tighter decision boundaries, which may be beneficial in certain tasks (Huang et al., 2020). This interpretation offers a useful lens through which to interpret our findings: improved safety performance does not require flatness, and may in some cases arise from sharper solutions.

Table 7: Results for ResNet18 (Pre-Trained) on TinyImageNet. Bolded values indicate the best performance per metric. For sharpness metrics, lower values correspond to flatter models.

| Control Condition | Generalisation Gap | Test Accuracy | Test ECE | Prediction Disagreement | Fisher Rao Norm | SAM Sharpness |
|---|---|---|---|---|---|---|
| Baseline | 49.643 $\pm$0.103 | 0.503 $\pm$0.001 | 0.257 $\pm$0.001 | 0.385 $\pm$0.000 | 0.479 $\pm$0.002 | 3.202E-04 $\pm 9.872E-06$ |
| Baseline + SAM | 46.255 $\pm$0.128 | 0.537 $\pm$0.001 | 0.223 $\pm$0.001 | 0.344 $\pm$0.000 | 0.427 $\pm$0.004 | 3.080E-04 $\pm 8.424E-06$ |
| Augmentation | 19.993 $\pm$0.091 | 0.508 $\pm$0.001 | 0.102 $\pm$0.001 | 0.544 $\pm$0.000 | 25.887 $\pm$0.098 | 1.680E+00 $\pm 8.776E-02$ |
| Augmentation + SAM | **16.777** $\pm$0.084 | 0.520 $\pm$0.001 | **0.044** $\pm$0.001 | 0.514 $\pm$0.000 | 25.193 $\pm$0.034 | 1.446E+00 $\pm 6.332E-02$ |
| Weight Decay | 49.689 $\pm$0.092 | 0.503 $\pm$0.001 | 0.202 $\pm$0.001 | 0.384 $\pm$0.000 | 0.998 $\pm$0.002 | 2.297E-04 $\pm 9.718E-06$ |
| Weight Decay+ SAM | 46.061 $\pm$0.111 | **0.539** $\pm$0.001 | 0.177 $\pm$0.001 | **0.339** $\pm$0.000 | 0.736 $\pm$0.004 | 3.784E-04 $\pm 9.996E-06$ |

**There is No Geometric Goldilocks Zone for Sharpness:** Although sharper solutions often perform better across generalisation and safety metrics on the CIFAR datasets, the sharpest model is not always the best overall. Still, the top-performing model is typically sharper than the Baseline, suggesting that a learning task may require a level of sharpness beyond what is induced by the architecture's implicit regularisation. This supports the view that neither extreme flatness nor sharpness is universally optimal. Instead, the "right" level of sharpness appears task- and architecture-dependent. This is also shown in Appendix Section H where we sweep the radius $\rho$ hyperparameter of SAM and observe that an increased radius leads to sharper models over the baseline that perform far better. However, in these experiments the sharpest model, when $\rho$ is 0.50, is not the best performing model. Importantly, this highlights the risk of misleading conclusions when aggregating sharpness trends across heterogeneous architectures: we observe that general trends can invert under such aggregation, consistent with Simpson's Paradox (Simpson, 1951). Careful control over architecture-specific inductive biases is therefore essential when studying geometry-function relationships.

## 7 CONCLUSION

This work revisits the relationship between geometry and generalisation in deep learning, extending it to include safety-relevant evaluations such as calibration, robustness to corruptions, and functional consistency. Rather than focusing solely on accuracy, we evaluate how sharpness relates to broader reliability properties. Across diverse architectures and datasets, we find that standard training controls such as weight decay, data augmentation, and SAM often lead to sharper solutions that also achieve stronger performance on safety metrics. These results challenge the conventional assumption that flatter minima are inherently preferable, and instead support a function-centric view in which sharper minima can correspond to more complex, well-generalising functions. We further reconcile SAM's behaviour by noting it promotes local robustness rather than global flatness, explaining why improved generalisation can coincide with increased sharpness. Our findings demonstrate that sharpness is not universally harmful – in fact, it may be beneficial for safety performance in certain settings. We posit that the geometry of learned solutions is shaped by task-specific demands, such as the need for tighter decision boundaries. Overall, this work calls for a re-evaluation of geometric intuitions in deep learning, and underscores the importance of connecting training controls, solution geometry, and functional reliability.

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

## A  SINGLE-OBJECTIVE OPTIMISATION FUNCTIONS

The Sphere, Rosenbrock, Rastrigin, Beale, Booth, Three Hump Camel and the Himmelblaus functions are defined in equations 2-8 respectively.

$$f(\boldsymbol{x}, \boldsymbol{y}) = (x^2 + y^2) \tag{2}$$

$$f(\boldsymbol{x}, \boldsymbol{y}) = (a - x)^2 + b(y - x^2)^2, \text{ where } a = 1 \text{ and } b = 100 \tag{3}$$

$$f(\boldsymbol{x}, \boldsymbol{y}) = 2a + x^2 - a\cos(2x\pi) + y^2 - a\cos(2y\pi), \text{ where } a = 10 \tag{4}$$

$$f(\boldsymbol{x}, \boldsymbol{y}) = (1.5 - x + xy)^2 + (2.25 - x + xy^2)^2 + (2.625 - x + xy^3)^2 \tag{5}$$

$$f(\boldsymbol{x}, \boldsymbol{y}) = (x + 2y - 7)^2 + (2x + y - 5)^2 \tag{6}$$

$$f(\boldsymbol{x}, \boldsymbol{y}) = 2x^2 - 1.05x^4 + \frac{x^6}{6} + xy + y^2 \tag{7}$$

$$f(\boldsymbol{x}, \boldsymbol{y}) = (x^2 + y - 11)^2 + (x + y^2 - 7)^2 \tag{8}$$

## A.1 TRAINING DETAILS

We trained a 3 layer ReLU multi-layered perceptron with a input width of two, a hidden width of 64 and output width of 1 with the Adam Optimizer with a learning rate of 1e-3. The train and test dataset consisted of 10,000 input pairs $(X, Y)$ generated by independently sampling $X$ and $Y$ from a uniform distribution $\mathcal{U}(-3.5, 3.5)$. For each of the seven functions (Sphere, Rosenbrock, Rastrigin, Beale, Booth, Three Hump Camel and the Himmelblaus), every input pair was evaluated using that specific function, yielding a target output $T$ for each function such that $F(X, Y) = T$. This procedure resulted in seven distinct datasets with identical input distributions but unique output transformations determined by their respective functions allowing for a clear assessment and comparison of the model's capacity to learn each target function under controlled input conditions.

For Figure 3 the model was trained 10 times with the same initialisation with ten different datasets for the respective function for $10^6$ epochs, where the mean sam sharpness based on the training data and train and test loss where recorded for initialisation and epochs $10^0, 10^1, 10^2, 10^3, 10^4, 10^5, 10^6$.

For Figures 5-9, the model was trained 10 times with the same initialisation with ten different datasets for the respective function until the mode the model reached the specified training target loss of $300, 150, 100, 10$, and $1$. For the Beale function, the model was unable to achieve a train loss of than 150 and lower within $10^6$ epochs, and for the Rosenbrock function the model was unable to achieve a train loss of 100 and lower within $10^6$ epochs.

## A.2 TRAINING TO EQUIVALENT LOSS

Because the model achieves different final losses after training for $[10^0, 10^1, 10^2, 10^3, 10^4, 10^5, 10^6]$ epochs, we control for training duration by fixing a target train loss. We then investigate how reaching an approximate target train loss influences model sharpness and the generalisation gap.

When comparing the mean sam sharpness a model achieves at a train loss of 300 (Figure 5), we observe clear patterns. The model trained on the Rosenbrock and Beale tasks has sharpness values between 20 and 50. In contrast, when trained on Rastrigin, Booth, and Himmelblaus tasks, sharpness values range between 5 and 10. The Sphere and Three-Hump Camel tasks produce the flattest results. In Figure 5(centre), the model trained on different tasks shows varied generalisation gaps at this fixed loss. In Figure 5(right), several tasks (Sphere, Rastrigin, Booth, Three-Hump Camel, Himmelblaus) yield similar generalisation gaps but differing sharpness values. Interestingly, the Rosenbrock task produces significantly higher sharpness while overlapping in generalisation gap with the Three-Hump Camel task. These observations underscore that sharpness reflects the learning task rather than model generalisation.

Because measuring at a train loss of 300 is arbitrary, we also examine target losses of 150, 100, 10, and 1 (Figures 6-9). Across these, we find that the model can have similar train losses but different sharpness depending on the learned function, supporting the initial claim.

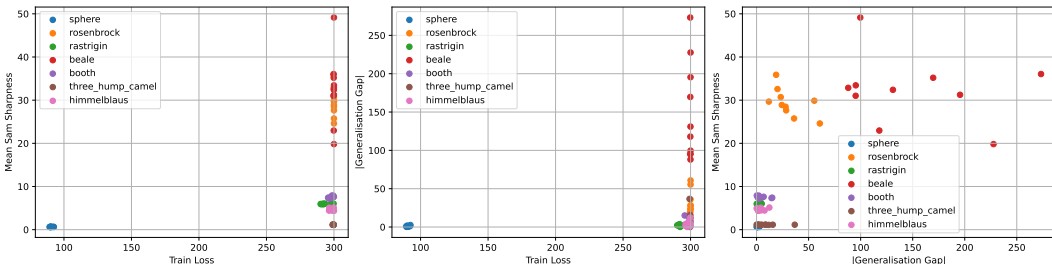

Figure 5: Scatter plots an MLP trained on the sphere, rosenbrock, rastrigin, beale, booth, three-hump camel, and himmelblaus functions for 10 different datasets till reaching a target train loss of 300: (left) mean sam sharpness vs. train loss, (centre) | generalisation gap | vs. train loss, and (right) |generalisation gap| vs. mean sam sharpness.

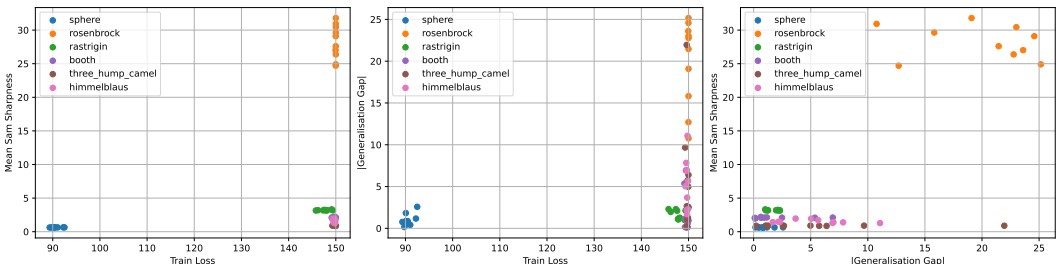

Figure 6: Scatter plots an MLP trained on the sphere, rosenbrock, rastrigin, booth, three-hump camel, and himmelblaus functions for 10 different datasets till reaching a target train loss of 150: (left) mean sam sharpness vs. train loss, (centre) | generalisation gap | vs. train loss, and (right) |generalisation gap| vs. mean sam sharpness.

Some functions drop off as we reach particular target losses. This happens because functions with more complicated landscapes, such as Beale and Rosenbrock, cannot exceed a train loss of 150 MSE. Less complex functions, such as the Sphere, can surpass this threshold. This supports our understanding that function complexity and solution geometry impact how easily a function can be fit. Less complex functions are more easily fit and tend to record lower sharpness values than complex functions, even when they achieve the same relative loss and generalisation gaps. As a result, it may be necessary to have better inductive biases for such complicated functions that are not captured under traditional initialisation strategies.

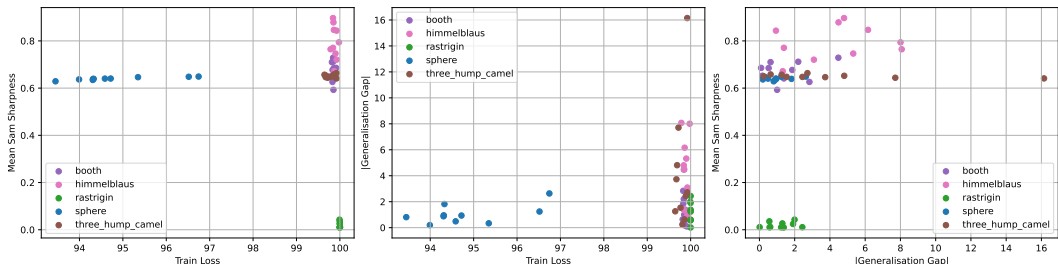

Figure 7: Scatter plots an MLP trained on the sphere, rastrigin, booth, three-hump camel, and himmelblaus functions for 10 different datasets till reaching a target train loss of 100: (left) mean sam sharpness vs. train loss, (centre) | generalisation gap | vs. train loss, and (right) |generalisation gap| vs. mean sam sharpness.

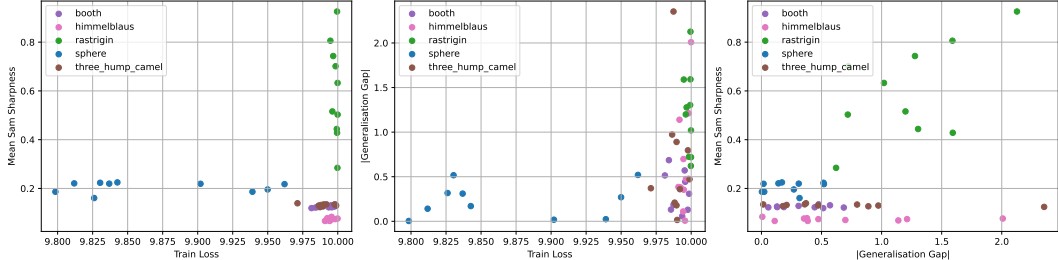

Figure 8: Scatter plots an MLP trained on the sphere,, booth, three-hump camel, and himmelblaus functions for 10 different datasets till reaching a target train loss of 10: (left) mean sam sharpness vs. train loss, (centre) | generalisation gap | vs. train loss, and (right) |generalisation gap| vs. mean sam sharpness.

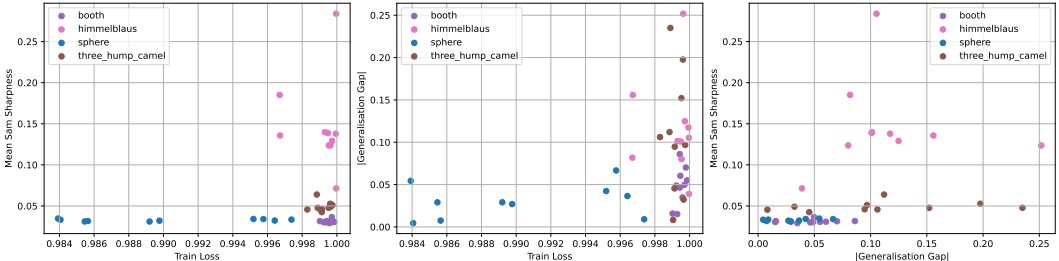

Figure 9: Scatter plots an MLP trained on the sphere,, booth, three-hump camel, and himmelblaus functions for 10 different datasets till reaching a target train loss of 1: (left) mean sam sharpness vs. train loss, (centre) | generalisation gap | vs. train loss, and (right) |generalisation gap| vs. mean sam sharpness.

## B  SHARPNESS METRICS

This section describes the sharpness metrics Fisher-Rao norm, SAM-Sharpness and Relative Flatness. Information Geometric Sharpness (IGS) (Jang et al., 2022) is also a suitable sharpness metric candidate, however we omitted it from this study as the calculation of this metric exceeds feasible computation for large-networks and dataset sizes. For implemenations of Fisher-Rao and Relative Flatness we use the code base provided by Petzka et al. (2021) [1].

**Fisher-Rao**  Fisher-Rao Norm (Liang et al., 2019) uses information Geometry for norm-based complexity measurement. It provides a reparametrisation invariant measure for loss landscape sharpness measuring, as verified by Petzka et al. (2021) in line with Petzka et al. (2021) we use the analytical formula for cross entropy loss from Appendix (Liang et al., 2019) which is presented in equation 9. To calculate the number of layers we sum the number of Linear, Conv1d, Conv2d, Conv3d and Embedding layers in a specified neural network for our experiments this means that the VGG19 has 18 layers, the ResNet18 has 21 and the ViT has 26.

$$\text{FR}_{\text{norm}} = \sqrt{(L+1)^2 \cdot \frac{1}{N} \sum_{i=1}^{N} \left( \frac{\partial \ell_i}{\partial \theta} \cdot \theta \right)} \qquad (9)$$

**SAM-Sharpness**  We define SAM-sharpness as the average difference across 100 different locations of $0.005\rho$ away the original model and calculate the SAM sharpness from these models as defined

---

[1]Code base for sharpness metrics Fisher Rao Norm and Relative Flatness from Petzka et al. (2021): https://github.com/kampmichael/RelativeFlatnessAndGeneralization/blob/main/CorrelationFlatnessGeneralization/measure_comparison.py

by Mason-Williams et al. (2024a) and Foret et al. (2021).

$$S(\theta) = \frac{1}{K} \sum_{k=1}^{K} \left| \frac{L(\theta + \Delta\theta_k) - L(\theta)}{\rho} \right|. \tag{10}$$

**Relative Flatness**  Petzka et al. (2021) define the sharpness measure Relative Flatness– their results show that it has the strongest correlational between flatness and a low generalisation gap. Relative Flatness sharpness is calculated between the feature extraction layer and the classification of the neural network and represents a highly expensive measure due to its calculation of the trace of the hessian of these output matrices. The formula for Relative Flatness from Han et al. (2025) can be found in equation 11.

$$\kappa_{\text{Tr}}^{\phi}(\mathbf{w}) := \sum_{s,s'=1}^{d} \langle \mathbf{w}_s, \mathbf{w}_{s'} \rangle \cdot \text{Tr}(H_{s,s'}(\mathbf{w}, \phi(S))) \tag{11}$$

where $w_s$ denotes the s-th row of $w$, $\langle \cdot, \cdot \rangle$ is the scalar product, and $H_{s,s'}(\mathbf{w}, \phi(S))$ is the Hessian of the empirical loss with respect to $w_s$ and $w_{s'}$ evaluated at $\phi(S)$ (Han et al., 2025).

## C  SAFETY CRITICAL METRICS

**Expected Calibration Error**  Calibration is the deviation of predicted confidence of a neural network and the true probabilities observed in the data, Guo et al. (2017) explored how ResNets are poorly calibrated and are often over confident. To calculate Expected Calibration Error (ECE) we use the Lighting AI Pytorch Metrics implementation of Multiclass Calibration Error[2] Implemented from Kumar et al. (2019).

$$\text{ECE} = \sum_{i=1}^{N} b_i \|p_i - c_i\|_1 \tag{12}$$

Where $p_i$ represents accuracy in bin $i$. The average confidence for predictions is $c_i$ in the bin with uniform sampling (Kumar et al., 2019)[2].

**Functional Diversity**  To provide an intuitive understanding of functional diversity we are interested the deviations between models top-1 predictions, the metric we focus on for this is: **Prediction Disagreement** which represents the disagreement between the top-1 predictions of two models on the test dataset as defined in equation 13 by Fort et al. (2020), where each $f(x; \theta)$ is the top-1 predicted class for a given sample $x$, operated on by parameters, $\theta$. A lower Prediction Disagreement results in a models that agree more on top-1 predictions.

$$\frac{1}{N} \sum_{n=1}^{N} \left[ f(x_n; \theta_1) \neq f(x_n; \theta_2) \right]. \tag{13}$$

**Robustness Evaluations**  We employ the CIFAR10-C and CIFAR100-C datasets provided by Hendrycks & Dietterich (2019) to observe how geometric properties interact with the robustness of a neural network. The corruptions have 5 levels of severity per perturbation.

**Corruption Accuracy (cACC)**  The metric we used for this robustness analysis is Corruption Accuracy. It represents the average accuracy of a classifier (f) on an average-case perturbed test dataset ($\mathcal{D}_{corruption}$) across permutation strengths 1-5 (Hendrycks & Dietterich, 2019).

$$\text{Corruption Accuracy} = \frac{1}{C} \sum_{c=1}^{C} \frac{1}{N_c} \sum_{n=1}^{N_c} 1(f(x_n; \theta) = y_n)). \tag{14}$$

---

[2]Calibration Error documentation from Lighting AI:https://lightning.ai/docs/torchmetrics/stable/classification/calibration_error.html

Where $C$ is number of corruptions, $N_c$, is the number of samples in corruption $c$, $f(x_n; \theta)$ represents the top 1 prediction of a class for a given sample $x$ with parameters $\theta$ and $y_n$ is the label.

## D  EXPERIMENTAL SETTINGS

All models are trained using NVIDIA A100 GPU's and each sharpness metric is calculated using the same GPU setup - as models output layer becomes larger for transitions between CIFAR10, CIFAR100 and TinyImageNet the computational cost of the calculation of sharpness metrics increases (by an order of magnitude between CIFAR10 and CIFAR100). It should be noted that while Fisher Rao Norm is computationally inexpensive to calculate, SAM sharpness takes a factor of time longer and Relative Flatness is the most computationally expensive measure from a time and memory perspective. All models are trained such that they converge on the training dataset or approximately converge in the case of augmentation conditions - it is important to note that all models are given **100 epochs to reduce loss on the training** set to make comparisons fair. As a result, the test error is appropriate for assessing the generalisation gap as a high test accuracy is indicative of a small generalisation gap.

**CIFAR10 Training:**  To train the **baseline** architectures on the CIFAR10 dataset we use the following settings: We use SGD with the momentum hyperparameter at 0.9 to minimize cross entropy loss for 100 epochs, using a batch size of 256 a learning rate of 0.001. For all architectures in the **SAM condition** we use the same settings as above but with SAM an extra optimization step occurs. We use SAM with the hyperparameter $\rho$ at the standard value of 0.05. For the **Augmentation condition** we use the Baseline conditions with the augmentations Random Crop with a padding of 4 and a fill of 128 alongside a Random Horizontal Flip with a probability of 0.5. Finally for the **Weight Decay condition** we use the same setup as the Baseline condition but with the addition of the weight decay value set at $5e^{-4}$.

**CIFAR10 Sharpness:**  For all sharpness metrics on CIFAR10 we used the entire training dataset to calculate sharpness across Fisher Rao Norm, SAM Sharpness and Relative Flatness. For the augmentation condition, the training dataset is the augmentations data used to train the model. We show in Sections E.1 and F.2 that calculating sharpness on the augmented training dataset for the models in the augmentation condition is approximately equivalent to calculating with the original training dataset without augmentation, thus preserving the same trends of increased sharpness for models trained with augmentation.

**CIFAR100 Training:**  To train the **baseline** architectures on the CIFAR100 dataset we use the following settings: We use SGD with the momentum hyperparameter at 0.9 to minimize cross entropy loss for 100 epochs, using a batch size of 256 a learning rate of $1e^{-2}$, we also use a Pytorch's (Paszke et al., 2019) Cosine Annealing learning rate scheduler with a Maximum number of iterations of 100. For all architectures in the **SAM condition** we use the same settings as above but with SAM as an extra optimization step occurs and for this we use SAM with the hyperparameter $\rho$ at the standard value of 0.05. For the **Augmentation condition** we use the Baseline conditions with the augmentations Random Crop with a padding of 4 and a fill of 128 alongside a Random Horizontal Flip with a probability of 0.5. Finally for the **Weight Decay condition** we use the same setup as the Baseline condition but with the addition of the weight decay value set at $5e^{-4}$.

**CIFAR100 Sharpness:**  For both the Fisher Rao Norm and SAM Sharpness metrics on CIFAR100 we used the entire training dataset to calculate sharpness. However, due to the computational burden of calculating Relative Flatness, we only employ 20% of the training dataset to calculate sharpness for this metrics. Once again, for the Augmentation condition, the training dataset is the augmentations data used to train the model.

**TinyImageNet Training:**  On the TinyImagenet dataset we use use pre-trained weights provided for the ResNet18 [3] and VGG19BN [4] by Pytorch - we modify these architectures by removing the

---

[3]Pytorch ResNet18 ImageNet1K Pretrained Model: `https://docs.pytorch.org/vision/main/models/generated/torchvision.models.resnet18.html`

[4]Pytorch VGG19BN ImageNet1K Pretrained Model: `https://docs.pytorch.org/vision/main/models/generated/torchvision.models.vgg19_bn.html`

existing final layer and replacing it with a final layer with a 200 output classification layer.

To train the **baseline** condition on these architectures using the following settings: We use SGD with the momentum hyperparameter at 0.9 to minimize cross entropy loss for 100 epochs, using a batch size of 256 a learning rate of 0.001. For all architectures in the **SAM condition** we use the same settings as above but with SAM as an extra optimization step occurs and for this we use SAM with the hyperparameter $\rho$ at the standard value of 0.05. For the **Augmentation condition** we use the Baseline conditions with the augmentations Random Resized Crop to the size of 64 and a Random Horizontal Flip with a probability of 0.5. Finally for the **Weight Decay condition** we use the same setup as the Baseline condition but with the addition of the weight decay value set at $5e^{-4}$.

**TinyImageNet Sharpness:** For the Fisher Rao Norm sharpness metric on TinyImageNet we used the entire training dataset to calculate sharpness. However, due to the computational burden of calculating SAM Sharpness, we only employ 20% of the training dataset to calculate sharpness for this metrics. Due to memory constraints on the A100 GPU's we were unable to calculate Relative Flatness for any size of the training dataset on this architecture. Once again, for the Augmentation condition, the training dataset is the augmentations data used to train the model.

# E    ResNet-18 Further Results

## E.1    Augmented or Standard Training Data Sharpness Calculation

We argue that the standard dataset is a subset of the augmented training dataset. Thus, sharpness trends are similar for both datasets. Our results show that calculating sharpness with augmented data is nearly identical to using the standard dataset for Fisher Rao Norm, Sam Sharpness, Relative Flatness, and loss landscape visualizations.

**Sharpness Metrics**    When calculating the sharpness metrics, it can be seen that the difference between using augmented training data, in Table 8, or standard training data, in Table 9, for each of the metrics provides no difference for the trends of results observed.

Table 8: Sharpness Calculation for ResNet18 landscape on CIFAR10 trained with batch size of 256 and learning rate of 0.001 using augmented training data for sharpness calculations.

| Control Condition | Fisher Rao Norm | SAM Sharpness | Relative Flatness |
|---|---|---|---|
| Augmentation | 3.940 $\pm$0.207 | 1.905E-01 $\pm 2.203E-02$ | 2903.220 $\pm$89.243 |
| Augmentation + SAM | 5.571 $\pm$0.035 | 1.303E-01 $\pm 1.547E-02$ | 4970.972 $\pm$30.139 |

Table 9: Sharpness Calculation for ResNet18 landscape on CIFAR10 trained with batch size of 256 and learning rate of 0.001 using standard training data for sharpness calculations.

| Control Condition | Fisher Rao Norm | SAM Sharpness | Relative Flatness |
|---|---|---|---|
| Augmentation | 3.962 $\pm$0.292 | 1.591E-02 $\pm 1.609E-03$ | 2972.554 $\pm$137.079 |
| Augmentation + SAM | 5.084 $\pm$0.032 | 2.035E-02 $\pm 1.203E-03$ | 5105.327 $\pm$43.058 |

**Loss Landscape Visualisations**    In Table 10, we show that the use of augmented or standard training data has little impact on the resulting loss landscape visualisation. This reaffirms that it is valid to calculate sharpness for models using augmented training data. The dataset used does not significantly impact the sharpness of the landscape or the resulting sharpness values. The standard dataset is simply a subset of the augmented data. Furthermore, sharpness calculation depends more on the model weights than on the data, and should be a representative value for any dataset given the same weight permutations.

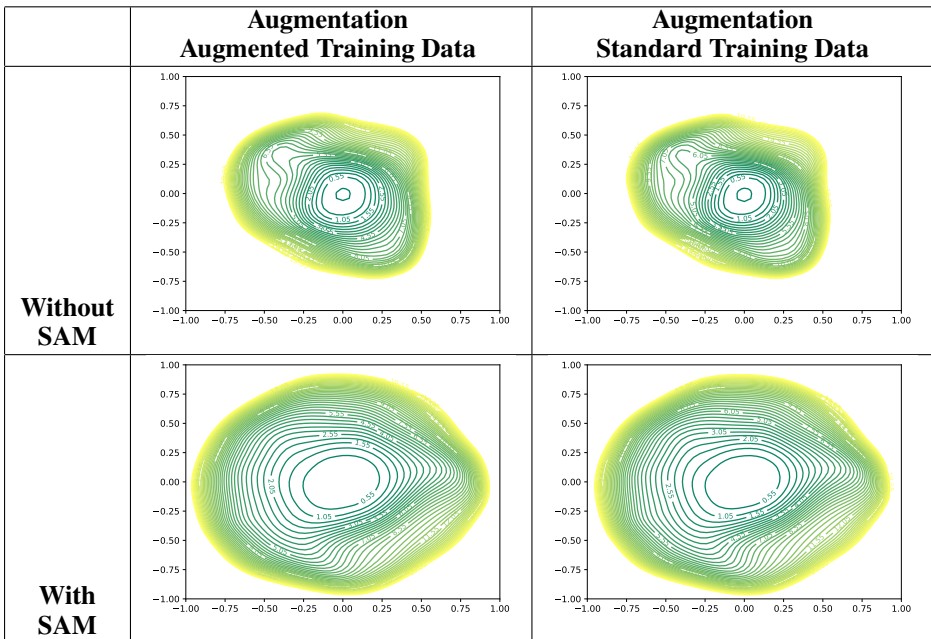

Table 10: Loss landscape visualisation (Li et al., 2018) of ResNet18 landscape on CIFAR10 exploring the loss in the domain of the perturbations $[1, 1]^2$ with 51 steps in both directions on models trained with augmentation visualising landscape with standard training data and augmented training data.

## E.2 RESNET-18 BATCH SIZE AND LEARNING RATE HYPERPARAMETER SWEEP

Here we observe how two core hyperparameters, batch size and learning rate impact the general finding that models under the use of training regularisation navigate to sharper points and thus tighter decision boundaries than base models without their application. In line with the findings in the main paper we observe that the best performing models in each condition are those that are sharper than the baseline models for each respective experimental up.

Table 11: Results for ResNet-18 Trained on CIFAR10 with **batch size 256 and a learning rate of $\mathbf{1e^{-3}}$**. Numbers in bold indicate best scores for metrics. For sharpness metrics lower values represent flatter models.

| Condition Condition | Generalisation Gap | Test Accuracy | Test ECE | Corruption Accuracy | Prediction Disagreement | Fisher Rao Norm | SAM Sharpness | Relative Flatness |
|---|---|---|---|---|---|---|---|---|
| Baseline | 28.050 ±0.175 | 0.720 ±0.002 | 0.186 ±0.001 | 58.614 ±0.201 | 0.282 ±0.001 | 0.032 ±0.001 | 1.366E-05 ±1.206$E-06$ | 34.607 ±0.757 |
| Baseline + SAM | 20.588 ±0.125 | 0.794 ±0.001 | 0.108 ±0.001 | 66.342 ±0.164 | 0.168 ±0.000 | 0.107 ±0.006 | 5.823E-05 ±9.056$E-06$ | 75.093 ±1.693 |
| Augmentation | 10.399 ±0.067 | 0.886 ±0.001 | 0.077 ±0.001 | 68.755 ±0.219 | 0.121 ±0.001 | 3.940 ±0.207 | 1.905E-01 ±2.203$E-02$ | 2903.220 ±89.243 |
| Augmentation + SAM | **6.864** ±0.038 | **0.908** ±0.000 | **0.014** ±0.001 | **71.419** ±0.283 | **0.069** ±0.000 | 5.571 ±0.035 | 1.303E-01 ±1.547$E-02$ | 4970.972 ±30.139 |
| Weight Decay | 27.942 ±0.196 | 0.721 ±0.002 | 0.174 ±0.002 | 58.562 ±0.227 | 0.281 ±0.001 | 0.065 ±0.004 | 3.391E-05 ±4.494$E-06$ | 59.767 ±3.009 |
| Weight Decay + SAM | 19.788 ±0.149 | 0.802 ±0.001 | 0.096 ±0.001 | 67.079 ±0.117 | 0.162 ±0.001 | 0.127 ±0.006 | 8.733E-05 ±1.430$E-05$ | 88.807 ±2.336 |

Table 12: Results for ResNet-18 Trained on CIFAR10 with **batch size 256 and a learning rate of $\mathbf{1e^{-2}}$**. Numbers in bold indicate best scores for metrics. For sharpness metrics lower values represent flatter models.

| Control Condition | Generalisation Gap | Test Accuracy | Test ECE | Corruption Accuracy | Prediction Disagreement | Fisher Rao Norm | SAM Sharpness | Relative Flatness |
|---|---|---|---|---|---|---|---|---|
| Baseline | 16.203 ±0.266 | 0.838 ±0.003 | 0.109 ±0.004 | 70.814 ±0.390 | 0.138 ±0.001 | 0.015 ±0.006 | 7.648E-06 ±3.794$E-06$ | 16.641 ±5.960 |
| Baseline + SAM | 14.549 ±0.059 | 0.855 ±0.001 | 0.084 ±0.001 | 72.618 ±0.161 | 0.110 ±0.000 | 0.042 ±0.002 | 2.019E-05 ±2.683$E-06$ | 49.022 ±1.901 |
| Augmentation | 7.593 ±0.092 | 0.921 ±0.001 | 0.056 ±0.003 | 72.923 ±0.223 | 0.078 ±0.000 | 2.390 ±0.268 | 1.091E-01 ±1.958$E-02$ | 1604.778 ±103.972 |
| Augmentation + SAM | **6.920** ±0.056 | **0.931** ±0.001 | **0.037** ±0.001 | **73.483** ±0.212 | **0.058** ±0.000 | 1.165 ±0.014 | 2.267E-02 ±2.070$E-03$ | 1173.090 ±15.607 |
| Weight Decay | 16.791 ±0.122 | 0.832 ±0.001 | 0.071 ±0.001 | 68.538 ±0.159 | 0.157 ±0.000 | 0.097 ±0.001 | 3.673E-05 ±4.662$E-06$ | 99.041 ±0.736 |
| Weight Decay+ SAM | 14.022 ±0.089 | 0.860 ±0.001 | 0.050 ±0.001 | 73.100 ±0.137 | 0.116 ±0.000 | 0.446 ±0.013 | 3.238E-04 ±6.384$E-05$ | 178.169 ±4.103 |

Table 13: Results for ResNet-18 Trained on CIFAR10 with **batch size 128 and a learning rate of** $\mathbf{1e^{-3}}$. Numbers in bold indicate best scores for metrics. For sharpness metrics lower values represent flatter models.

| Control Condition | Generalisation Gap | Test Accuracy | Test ECE | Corruption Accuracy | Prediction Disagreement | Fisher Rao Norm | SAM Sharpness | Relative Flatness |
|---|---|---|---|---|---|---|---|---|
| Baseline | 23.325 ±0.140 | 0.767 ±0.001 | 0.154 ±0.001 | 63.035 ±0.204 | 0.227 ±0.000 | 0.013 ±0.000 | 5.181E-06 ±5.856$E-07$ | 27.916 ±0.340 |
| Baseline + SAM | 16.714 ±0.125 | 0.833 ±0.001 | 0.083 ±0.001 | 69.769 ±0.108 | 0.126 ±0.000 | 0.072 ±0.006 | 2.640E-05 ±4.345$E-06$ | 139.589 ±2.679 |
| Augmentation | 9.110 ±0.079 | 0.905 ±0.001 | 0.065 ±0.001 | 71.516 ±0.308 | 0.099 ±0.000 | 2.465 ±0.105 | 9.266E-02 ±5.276$E-03$ | 3735.018 ±173.247 |
| Augmentation + SAM | **6.869** ±0.022 | **0.921** ±0.000 | **0.013** ±0.000 | **72.870** ±0.207 | **0.058** ±0.000 | 4.070 ±0.027 | 8.913E-02 ±8.054$E-03$ | 7532.582 ±69.191 |
| Weight Decay | 23.504 ±0.136 | 0.765 ±0.001 | 0.137 ±0.001 | 62.879 ±0.214 | 0.231 ±0.000 | 0.047 ±0.000 | 2.241E-05 ±3.426$E-06$ | 80.599 ±0.548 |
| Weight Decay + SAM | 16.433 ±0.096 | 0.836 ±0.001 | 0.072 ±0.001 | 70.226 ±0.158 | 0.124 ±0.000 | 0.110 ±0.004 | 4.797E-05 ±7.285$E-06$ | 194.034 ±2.840 |

Table 14: Results for ResNet-18 Trained on CIFAR10 with **batch size 128 and a learning rate of** $\mathbf{1e^{-2}}$. Numbers in bold indicate best scores for metrics. For sharpness metrics lower values represent flatter models.

| Control Condition | Generalisation Gap | Test Accuracy | Test ECE | Corruption Accuracy | Prediction Disagreement | Fisher Rao Norm | SAM Sharpness | Relative Flatness |
|---|---|---|---|---|---|---|---|---|
| Baseline | 15.027 ±0.069 | 0.850 ±0.001 | 0.109 ±0.001 | 71.960 ±0.158 | 0.125 ±0.000 | 0.003 ±0.000 | 1.094E-06 ±9.618$E-08$ | 8.785 ±0.142 |
| Baseline + SAM | 13.231 ±0.065 | 0.868 ±0.001 | 0.081 ±0.001 | 73.053 ±0.164 | 0.099 ±0.000 | 0.024 ±0.001 | 1.021E-05 ±5.519$E-07$ | 70.694 ±2.273 |
| Augmentation | 7.455 ±0.062 | 0.923 ±0.001 | 0.057 ±0.001 | 72.594 ±0.152 | 0.076 ±0.000 | 2.086 ±0.140 | 8.274E-02 ±6.784$E-03$ | 2864.657 ±151.088 |
| Augmentation + SAM | **6.678** ±0.060 | **0.933** ±0.001 | **0.036** ±0.001 | **73.565** ±0.245 | **0.056** ±0.000 | 1.012 ±0.014 | 2.173E-02 ±2.042$E-03$ | 2354.005 ±38.058 |
| Weight Decay | 12.695 ±0.072 | 0.873 ±0.001 | 0.057 ±0.001 | 70.979 ±0.124 | 0.103 ±0.000 | 0.159 ±0.003 | 1.265E-04 ±3.140$E-06$ | 355.345 ±12.866 |
| Weight Decay + SAM | 12.606 ±0.069 | 0.874 ±0.001 | **0.036** ±0.001 | 72.795 ±0.162 | 0.107 ±0.000 | 0.745 ±0.017 | 5.880E-04 ±6.127$E-05$ | 439.467 ±8.439 |

# F  VGG-19

## F.1  VGG19 Batch Size and Learning Rate Hyperparameter Sweep

Here we observe how two core hyperparameters, batch size and learning rate, impact the general finding that models under the use of training regularisation navigate to sharper points and thus tighter decision boundaries than base models without regularisation. In line with the findings in the main paper, we observe that the best-performing models in each condition are those that are sharper than the baseline models for each respective experimental setup. However, it is important to note that modifying the learning rate and batch size does influence the sharpness values that we observe in each condition, with a larger learning rate typically increasing the flatness of the minima considerably more than using a smaller learning rate. However, within these augmented models still navigate to sharper landscapes than the baseline and achieve the best performance across generalization and safety evaluations.

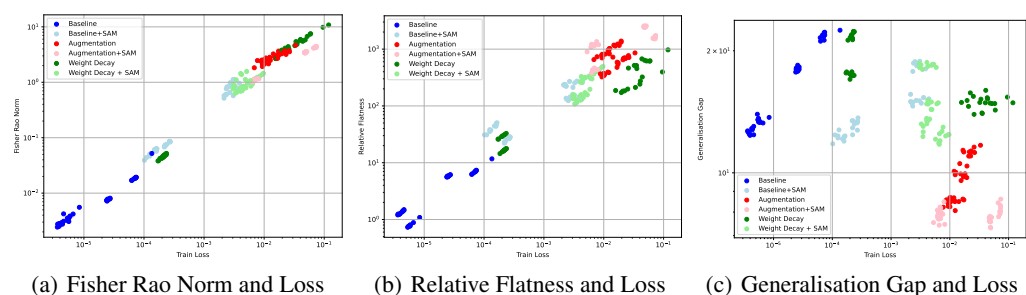

(a) Fisher Rao Norm and Loss    (b) Relative Flatness and Loss    (c) Generalisation Gap and Loss

Figure 10: Plot of 240 minima using reparametrisation invariant sharpness metrics against train loss and generalisation gap against train loss using log-scale for the VGG19 with different training hyperparameters (batch size of 256, 128 and learning rate of 0.001 and $1e^{-2}$) trained on CIFAR10.

**CIFAR10:**  The Augmentation and SAM condition perform the best for all metrics. It is also the sharpest model with the highest values for Relative Flatness and the second highest for SAM

Sharpness and Fisher Rao Norm value. These findings are consistent across the hyper parameter sweep that we perform across learning rate and batch size.

Table 15: Results for VGG19 Trained on CIFAR10 with **batch size 256 and a learning rate of** $1\mathrm{e}^{-3}$. Numbers in bold indicate best scores for metrics. For sharpness metrics lower values represent flatter models.

| Control Condition | Genralisation Gap | Test Accuracy | Test ECE | Corruption Accuracy | Prediction Disagreement | Fisher Rao Norm | SAM Sharpness | Relative Flatness |
|---|---|---|---|---|---|---|---|---|
| Baseline | 21.805 ±0.128 | 0.782 ±0.001 | 0.160 ±0.001 | 64.316 ±0.193 | 0.204 ±0.000 | 0.022 ±0.003 | 7.649E-06 ±2.207$E-06$ | 7.374 ±0.470 |
| Baseline + SAM | 18.444 ±0.097 | 0.815 ±0.001 | 0.108 ±0.001 | 66.655 ±0.296 | 0.150 ±0.000 | 0.938 ±0.036 | 1.495E-03 ±1.703$E-04$ | 140.164 ±3.149 |
| Augmentation | 11.289 ±0.066 | 0.879 ±0.001 | 0.084 ±0.001 | 68.497 ±0.199 | 0.121 ±0.000 | 3.505 ±0.155 | 1.967E-01 ±2.298$E-02$ | 688.897 ±26.348 |
| Augmentation + SAM | **8.139** ±0.074 | **0.903** ±0.001 | **0.019** ±0.001 | **71.268** ±0.196 | **0.075** ±0.000 | 4.278 ±0.027 | 9.777E-02 ±1.126$E-02$ | 1609.212 ±22.719 |
| Weight Decay | 21.801 ±0.121 | 0.782 ±0.001 | 0.151 ±0.001 | 64.405 ±0.217 | 0.202 ±0.000 | 0.048 ±0.001 | 1.315E-05 ±1.143$E-06$ | 16.494 ±0.292 |
| Weight Decay + SAM | 18.394 ±0.067 | 0.816 ±0.001 | 0.104 ±0.001 | 66.827 ±0.286 | 0.151 ±0.000 | 1.121 ±0.080 | 3.210E-03 ±6.174$E-04$ | 157.592 ±5.360 |

Table 16: Results for VGG19 Trained on CIFAR10 with **batch size 256 and a learning rate of** $1\mathrm{e}^{-2}$. Numbers in bold indicate best scores for metrics. For sharpness metrics lower values represent flatter models.

| Control Condition | Generalisation Gap | Test Accuracy | Test ECE | Corruption Accuracy | Prediction Disagreement | Fisher Rao Norm | SAM Sharpness | Relative Flatness |
|---|---|---|---|---|---|---|---|---|
| Baseline | 13.507 ±0.063 | 0.865 ±0.001 | 0.105 ±0.001 | 71.476 ±0.125 | 0.119 ±0.000 | 0.004 ±0.000 | 1.604E-06 ±1.637$E-07$ | 0.807 ±0.032 |
| Baseline + SAM | 13.183 ±0.115 | 0.868 ±0.001 | 0.081 ±0.001 | 71.908 ±0.121 | 0.103 ±0.000 | 0.077 ±0.002 | 4.290E-05 ±5.280$E-06$ | 25.287 ±0.720 |
| Augmentation | 8.565 ±0.030 | 0.910 ±0.000 | 0.065 ±0.000 | 71.491 ±0.442 | 0.092 ±0.000 | 2.555 ±0.154 | 1.146E-01 ±9.917$E-03$ | 396.136 ±22.479 |
| Augmentation + SAM | **7.969** ±0.080 | **0.920** ±0.001 | **0.040** ±0.001 | **73.037** ±0.160 | **0.071** ±0.000 | 1.087 ±0.014 | 3.122E-02 ±3.347$E-03$ | 429.679 ±6.763 |
| Weight Decay | 15.241 ±0.124 | 0.836 ±0.003 | 0.118 ±0.002 | 68.294 ±0.390 | 0.184 ±0.002 | 5.120 ±0.586 | 2.302E-02 ±4.987$E-03$ | 237.081 ±22.113 |
| Weight Decay + SAM | 13.188 ±0.130 | 0.868 ±0.001 | 0.063 ±0.001 | 71.618 ±0.284 | 0.117 ±0.000 | 0.742 ±0.022 | 5.764E-04 ±5.989$E-05$ | 124.537 ±3.152 |

Table 17: Results for VGG19 Trained on CIFAR10 with **batch size 128 and a learning rate of** $1\mathrm{e}^{-3}$. Numbers in bold indicate best scores for metrics. For sharpness metrics lower values represent flatter models.

| Control Condition | Generalisation Gap | Test Accuracy | Test ECE | Corruption Accuracy | Prediction Disagreement | Fisher Rao Norm | SAM Sharpness | Relative Flatness |
|---|---|---|---|---|---|---|---|---|
| Baseline | 18.026 ±0.066 | 0.820 ±0.001 | 0.140 ±0.001 | 68.325 ±0.110 | 0.167 ±0.000 | 0.008 ±0.000 | 8.460E-07 ±5.355$E-08$ | 5.824 ±0.058 |
| Baseline + SAM | 15.059 ±0.085 | 0.849 ±0.001 | 0.089 ±0.001 | 69.791 ±0.174 | 0.115 ±0.000 | 0.649 ±0.032 | 5.689E-04 ±1.248$E-04$ | 242.483 ±6.334 |
| Augmentation | 9.988 ±0.088 | 0.895 ±0.001 | 0.074 ±0.001 | 70.671 ±0.288 | 0.107 ±0.000 | 2.851 ±0.121 | 1.708E-01 ±1.967$E-02$ | 1158.004 ±40.307 |
| Augmentation + SAM | **7.594** ±0.050 | **0.916** ±0.000 | **0.017** ±0.000 | **72.194** ±0.175 | **0.066** ±0.000 | 3.469 ±0.023 | 7.664E-02 ±1.149$E-02$ | 2487.050 ±14.772 |
| Weight Decay | 17.485 ±0.066 | 0.825 ±0.001 | 0.127 ±0.001 | 68.655 ±0.156 | 0.158 ±0.000 | 0.044 ±0.001 | 4.918E-06 ±6.710$E-07$ | 29.566 ±0.647 |
| Weight Decay + SAM | 14.803 ±0.082 | 0.851 ±0.001 | 0.082 ±0.001 | 69.699 ±0.181 | 0.114 ±0.000 | 0.802 ±0.022 | 1.089E-03 ±3.140$E-04$ | 298.386 ±7.049 |

Table 18: Results for VGG19 Trained on CIFAR10 with **batch size 128 and a learning rate of** $1\mathrm{e}^{-2}$. Numbers in bold indicate best scores for metrics. For sharpness metrics lower values represent flatter models.

| Control Condition | Generalisation Gap | Test Accuracy | Test ECE | Corruption Accuracy | Prediction Disagreement | Fisher Rao Norm | SAM Sharpness | Relative Flatness |
|---|---|---|---|---|---|---|---|---|
| Baseline | 12.661 ±0.066 | 0.873 ±0.001 | 0.100 ±0.001 | 71.689 ±0.113 | 0.111 ±0.000 | 0.003 ±0.000 | 3.227E-07 ±3.198$E-08$ | 1.331 ±0.031 |
| Baseline + SAM | 12.248 ±0.078 | 0.878 ±0.001 | 0.077 ±0.001 | 71.838 ±0.256 | 0.100 ±0.000 | 0.054 ±0.003 | 8.424E-06 ±9.410$E-07$ | 40.853 ±2.177 |
| Augmentation | 8.333 ±0.060 | 0.913 ±0.001 | 0.063 ±0.001 | 72.114 ±0.259 | 0.091 ±0.000 | 2.510 ±0.122 | 1.605E-01 ±2.042$E-02$ | 772.205 ±19.954 |
| Augmentation + SAM | **7.791** ±0.072 | **0.922** ±0.001 | **0.038** ±0.001 | **72.470** ±0.190 | **0.072** ±0.000 | 1.076 ±0.019 | 3.497E-02 ±4.347$E-03$ | 1155.243 ±35.334 |
| Weight Decay | 14.589 ±0.143 | 0.839 ±0.003 | 0.110 ±0.002 | 66.692 ±0.603 | 0.187 ±0.001 | 6.253 ±0.596 | 7.018E-03 ±8.372$E-04$ | 608.657 ±44.645 |
| Weight Decay + SAM | 12.295 ±0.112 | 0.877 ±0.001 | 0.052 ±0.001 | 71.167 ±0.301 | 0.116 ±0.001 | 1.193 ±0.043 | 5.407E-04 ±1.109$E-04$ | 406.163 ±15.541 |

**CIFAR10 Landscape Visualisation:**    Here we observe that the loss landscapes show that the use of regularisation does change the function learned by the model and that this can often increase in complexity. For example, in Table 19 we can see that the use of weight decay, augmentation and SAM all change the minima that is reached at the end of training, with weight decay and augmentation showing a big increase in complexity compared to the baseline landscape.

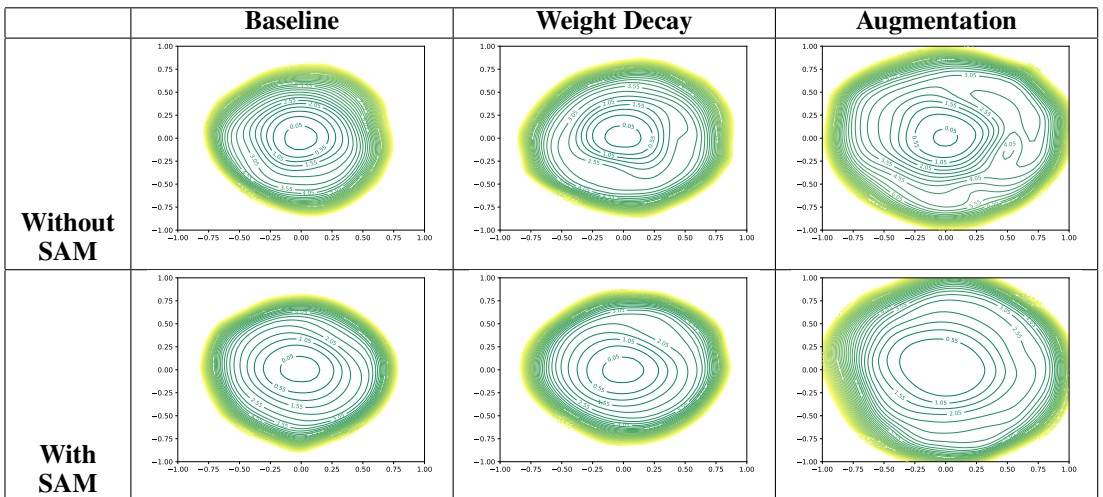

| | **Baseline** | **Weight Decay** | **Augmentation** |
|---|---|---|---|
| **Without SAM** | | | |
| **With SAM** | | | |

Table 19: Loss landscape visualisation (Li et al., 2018) of VGG19 landscape on CIFAR10 exploring the loss in the domain of the perturbations $[1, 1]^2$ with 51 steps in both directions.

### F.2 AUGMENTED OR STANDARD TRAINING DATA SHARPNESS CALCULATION

**Sharpness Metrics** When calculating the sharpness metrics, it can be seen that the difference between using augmented training data, in Table 20, or standard training data, in Table 21, for each of the metrics provides no difference for the trends of results observed, training with augmentation and augmentation + SAM results in a minima that is substantially sharper than a baseline model.

Table 20: Sharpness Calculation for VGG19 on CIFAR10 trained with batch size of 256 and learning rate of 0.001 using augmented training data for sharpness calculations.

| Control Condition | Fisher Rao Norm | SAM Sharpness | Relative Flatness |
|---|---|---|---|
| Augmentation | 3.505 ±0.155 | 1.967E-01 ±2.298E-02 | 688.897 ±26.348 |
| Augmentation + SAM | 4.278 ±0.027 | 9.777E-02 ±1.126E-02 | 1609.212 ±22.719 |

Table 21: Sharpness Calculation for VGG19 on CIFAR10 trained with batch size of 256 and learning rate of 0.001 using standard training data for sharpness calculations.

| Control Condition | Fisher Rao Norm | SAM Sharpness | Relative Flatness |
|---|---|---|---|
| Augmentation | 2.322 ±0.125 | 9.589E-03 ±1.438E-03 | 481.312 ±23.358 |
| Augmentation + SAM | 3.756 ±0.030 | 1.497E-02 ±7.363E-04 | 1413.712 ±22.212 |

**Loss Landscape Visualisations** In Table 22, we show that the use of augmented or standard training data has little impact on the resulting loss landscape visualisation. This reaffirms that it is valid to calculate sharpness for models using augmented training data. The dataset used does not significantly impact the sharpness of the landscape or the resulting sharpness values. The standard dataset is simply a subset of the augmented data. Furthermore, sharpness calculation depends more on the model weights than on the data, and should be a representative value for any dataset given the same weight permutations.

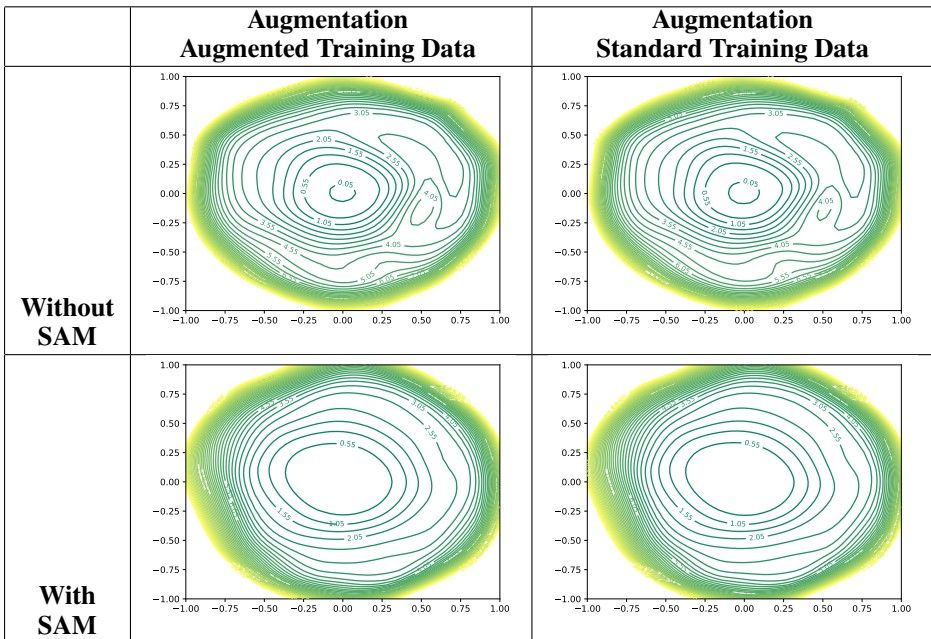

Table 22: Loss landscape visualisation (Li et al., 2018) of VGG19 landscape on CIFAR10 exploring the loss in the domain of the perturbations $[1,1]^2$ with 51 steps in both directions on models trained with augmentation visualising landscape with standard training data and augmented training data.

### F.3 CIFAR100:

Augmentation and SAM condition performs the best for test accuracy, Corruption Accuracy and Prediction Disagreement. However, for ECE we see that Weight Decay is the best condition. Augmentation and SAM is the second sharpest model for Fisher Rao Norm and SAM sharpness and has the highest value for Relative Flatness. It is important to note that for Weight Decay, with the lowest ECE, that it has higher sharpness values than the Baseline condition.

Table 23: Results for VGG-19 Trained on CIFAR100, the Mean and $\pm$ 1 SEM are recorded over 10 models. Numbers in bold indicate best scores for metrics. For sharpness metrics lower values represent flatter models.

| Control Condition | Generalisation Gap | Test Accuracy | Test ECE | Corruption Accuracy | Prediction Disagreement | Fisher Rao Norm | SAM Sharpness | Relative Flatness |
|---|---|---|---|---|---|---|---|---|
| Baseline | 42.454 ±0.092 | 0.575 ±0.001 | 0.253 ±0.000 | 40.749 ±0.124 | 0.396 ±0.000 | 0.158 ±0.017 | 2.123E-04 ±2.649$E-05$ | 8.384 ±0.151 |
| Baseline + SAM | 43.815 ±0.224 | 0.561 ±0.002 | 0.232 ±0.002 | 39.690 ±0.196 | 0.399 ±0.001 | 0.529 ±0.017 | 7.520E-04 ±5.791$E-05$ | 67.485 ±1.802 |
| Augmentation | 32.519 ±0.156 | 0.646 ±0.002 | 0.222 ±0.002 | 40.832 ±0.321 | 0.358 ±0.001 | 7.156 ±0.270 | 2.835E-01 ±1.439$E-02$ | 1430.826 ±53.977 |
| Augmentation + SAM | **32.008** ±0.099 | **0.656** ±0.001 | 0.157 ±0.001 | **41.276** ±0.089 | **0.326** ±0.001 | 5.653 ±0.073 | 1.971E-01 ±1.170$E-02$ | 2085.080 ±31.648 |
| Weight Decay | 41.579 ±0.107 | 0.584 ±0.001 | **0.138** ±0.000 | 41.266 ±0.112 | 0.384 ±0.000 | 0.678 ±0.008 | 3.302E-04 ±3.256$E-05$ | 45.728 ±0.073 |
| Weight Decay + SAM | 44.631 ±0.228 | 0.553 ±0.002 | 0.189 ±0.002 | 38.961 ±0.191 | 0.429 ±0.001 | 2.138 ±0.084 | 2.630E-03 ±2.111$E-04$ | 153.194 ±6.495 |

**CIFAR100 Landscape Visualisation:** Once again, we confirm through the loss landscape visualisation in Table 24, that the application of regularisers does indeed change the properties of the minima that a network reaches at the end of training. This, corroborates our findings that state that regularisation can change the function complexity of a network and thus impact the geometric properties of the minima found at the end of training.

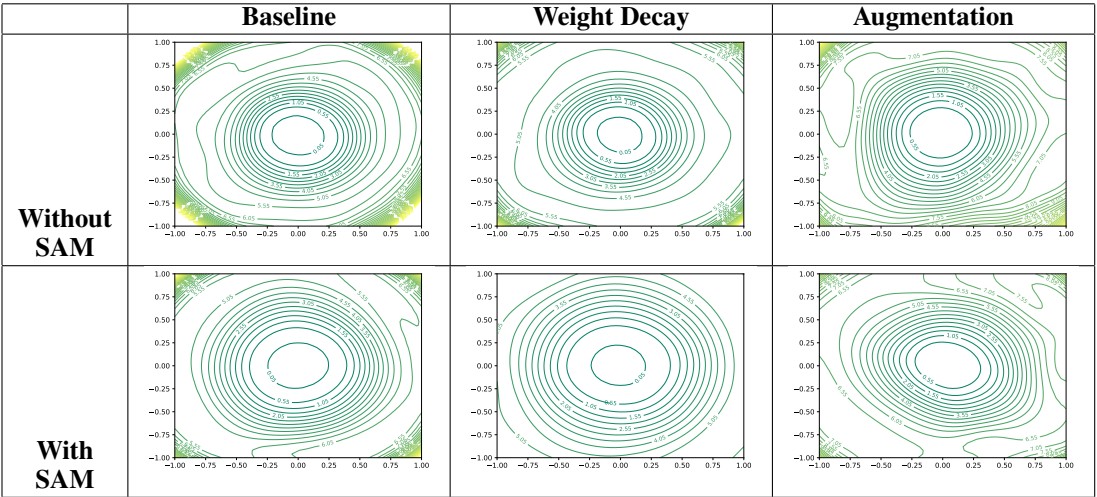

Table 24: Loss landscape visualisation (Li et al., 2018) VGG19 landscape on CIFAR100 exploring the loss in the domain of the perturbations $[1, 1]^2$ with 51 steps in both directions.

## F.4 TINYIMAGENET:

The Weight Decay and SAM condition performs best for test accuracy and Prediction Disagreement. For Weight Decay and SAM condition we see no real difference in the sharpness values. For ECE we see that Augmentation + SAM is the best condition. Augmentation and SAM is the second sharpest model for Fisher Rao Norm and SAM sharpness.

Table 25: Results for VGG19-BN (Pre-Trained) on TinyImageNet. Numbers in bold indicate best scores for metrics. For sharpness metrics lower values represent flatter models.

| Control Condition | Generalisation Gap | Test Accuracy | Test ECE | Prediction Disagreement | Fisher Rao Norm | SAM Sharpness |
|---|---|---|---|---|---|---|
| Baseline | 39.588 ±0.063 | 0.604 ±0.001 | 0.303 ±0.001 | 0.238 ±0.000 | 0.337 ±0.118 | 2.642E-04 ±$1.986E - 05$ |
| Baseline + SAM | 36.131 ±0.048 | 0.638 ±0.000 | 0.199 ±0.001 | 0.186 ±0.000 | 0.419 ±0.097 | 3.364E-04 ±$2.684E - 05$ |
| Augmentation | 20.952 ±0.080 | 0.578 ±0.001 | 0.119 ±0.001 | 0.473 ±0.000 | 20.033 ±0.076 | 1.893E+00 ±$7.702E - 02$ |
| Augmentation + SAM | **17.927** ±0.048 | 0.594 ±0.000 | **0.056** ±0.002 | 0.440 ±0.000 | 19.230 ±0.035 | 1.665E+00 ±$5.137E - 02$ |
| Weight Decay | 39.622 ±0.069 | 0.604 ±0.001 | 0.265 ±0.000 | 0.222 ±0.000 | 0.207 ±0.026 | 2.679E-04 ±$1.084E - 05$ |
| Weight Decay+ SAM | 35.922 ±0.050 | **0.641** ±0.001 | 0.180 ±0.001 | **0.185** ±0.000 | 0.342 ±0.015 | 3.072E-04 ±$4.621E - 06$ |

## G VISION TRANSFORMER

### G.1 CIFAR10

We see Augmentation and the Augmentation + SAM conditions perform best and they have the highest sharpness values across metrics.

Table 26: Results for ViT Trained on CIFAR10, the Mean and ± 1 SEM are recorded over 10 models. Numbers in bold indicate best scores for metrics. For sharpness metrics lower values represent flatter models.

| Control Condition | Generalisation Gap | Test Accuracy | Test ECE | Corruption Accuracy | Prediction Disagreement | Fisher Rao Norm | SAM Sharpness | Relative Flatness |
|---|---|---|---|---|---|---|---|---|
| Baseline | 39.040 ±0.177 | 0.610 ±0.002 | 0.308 ±0.002 | 54.805 ±0.147 | 0.408 ±0.001 | 0.221 ±0.003 | 8.769E-05 ±$4.974E - 06$ | 347.198 ±6.425 |
| Baseline + SAM | 39.935 ±0.144 | 0.600 ±0.001 | 0.276 ±0.001 | 54.792 ±0.113 | 0.421 ±0.001 | 1.576 ±0.083 | 1.458E-03 ±$8.995E - 05$ | 1459.292 ±82.220 |
| Augmentation | 1.305 ±0.076 | **0.724** ±0.001 | **0.019** ±0.001 | **64.092** ±0.152 | 0.217 ±0.001 | 22.809 ±0.117 | 4.741E-01 ±$3.822E - 02$ | 38465.647 ±139.905 |
| Augmentation + SAM | **-1.199** ±0.097 | 0.668 ±0.002 | 0.030 ±0.001 | 60.535 ±0.179 | **0.201** ±0.001 | 22.372 ±0.042 | 4.352E-01 ±$2.420E - 02$ | 18412.664 ±617.822 |
| Weight Decay | 38.746 ±0.196 | 0.613 ±0.002 | 0.301 ±0.002 | 55.077 ±0.159 | 0.402 ±0.001 | 0.328 ±0.003 | 1.359E-04 ±$1.030E - 05$ | 422.966 ±6.897 |
| Weight Decay + SAM | 39.881 ±0.162 | 0.600 ±0.002 | 0.268 ±0.001 | 54.797 ±0.125 | 0.419 ±0.001 | 2.250 ±0.099 | 2.890E-03 ±$3.102E - 04$ | 1908.688 ±97.800 |

## G.2 CIFAR100

We see Augmentation and the Augmentation + SAM conditions perform best and they have the highest sharpness values across metrics.

Table 27: Results for ViT Trained on CIFAR100, the Mean and $\pm$ 1 SEM are recorded over 10 models. Numbers in bold indicate best scores for metrics. For sharpness metrics lower values represent flatter models.

| Control Condition | Generalisation Gap | Test Accuracy | Test ECE | Corruption Accuracy | Prediction Disagreement | Fisher Rao Norm | SAM Sharpness | Relative Flatness |
|---|---|---|---|---|---|---|---|---|
| Baseline | 69.048 ±0.164 | 0.309 ±0.002 | 0.402 ±0.002 | 25.936 ±0.088 | 0.723 ±0.000 | 0.646 ±0.061 | 3.428E-04 ±4.954E − 05 | 112.185 ±4.246 |
| Baseline + SAM | 67.376 ±0.126 | 0.326 ±0.001 | 0.386 ±0.001 | 27.628 ±0.097 | 0.697 ±0.000 | 0.821 ±0.070 | 4.539E-04 ±6.066E − 05 | 124.472 ±30.314 |
| Augmentation | 37.472 ±0.249 | 0.508 ±0.001 | 0.227 ±0.001 | 38.680 ±0.091 | 0.483 ±0.001 | 17.321 ±0.192 | 5.995E-01 ±8.815E − 02 | 17401.462 ±143.009 |
| Augmentation + SAM | **32.136** ±0.262 | **0.523** ±0.001 | **0.146** ±0.002 | **40.275** ±0.097 | **0.446** ±0.000 | 19.664 ±0.127 | 4.649E-01 ±2.505E − 02 | 17812.985 ±55.523 |
| Weight Decay | 67.524 ±0.148 | 0.325 ±0.001 | 0.324 ±0.001 | 27.364 ±0.103 | 0.700 ±0.000 | 1.563 ±0.073 | 8.440E-04 ±1.251E − 04 | 251.148 ±15.330 |
| Weight Decay + SAM | 67.227 ±0.077 | 0.327 ±0.001 | 0.284 ±0.001 | 27.739 ±0.069 | 0.695 ±0.001 | 5.181 ±0.260 | 4.323E-03 ±3.837E − 04 | 1554.595 ±91.649 |

## H RADIUS ($\rho$) HYPERPARAMETER SWEEP FOR SAM SHARPNESS

We show that our finding of sharpness increasing under the application of SAM is robust to perturbations of the $\rho$ hyperparameter. We employ the $\rho$ value across the following values $0.5, 0.25, 0.05, 0.025, 0.005, 0.0025$ training the ResNet-18 with a **batch size 256 and a learning rate of $1\text{e}^{-3}$**. As shown in Figure 11, we can see that increasing the value of the $\rho$ hyperparameter increases the sharpness of the minima found at the end of training, which coincides with a reduced generalisation gap. Table 28, shows that when using a $\rho$ value of 0.25, we record the best accuracy, calibration, robustness and functional similarity results - coinciding with this finding, we can observe that this condition is far sharper than the other the other $\rho$ value below this, showing a relationship between increased sharpness and describable generalisation properties. Finally, it is important to note that the sharpest condition, found under a $\rho$ of 0.50, is not the best model. This reaffirms our understanding that the sharpness required to fit a function is highly dependant on the problem itself and that there appears to lack of a goldilocks zone of sharpness that is sufficient to fit a problem.

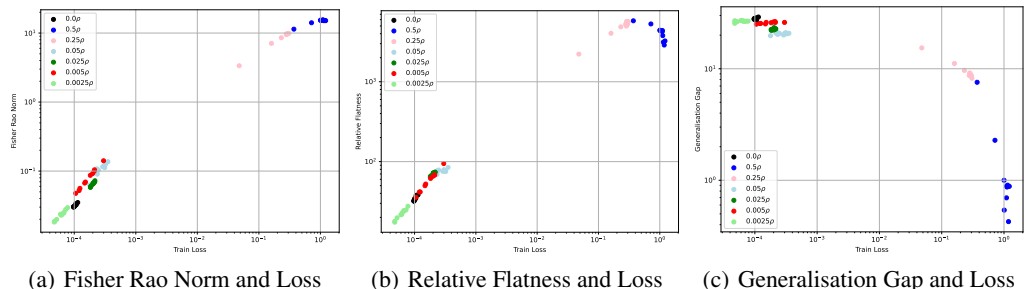

(a) Fisher Rao Norm and Loss    (b) Relative Flatness and Loss    (c) Generalisation Gap and Loss

Figure 11: Scatter plots of 60 converged minima for ResNet-18 on CIFAR-10 varying the SAM ($\rho$) hyperparameter (0.5, 0.25, 0.05, 0.025, 0.005, 0.0025) using batch size 256 and learning rate $10^{-3}$: (a) Fisher–Rao norm vs. train loss, (b) Relative Flatness vs. train loss, and (c) generalisation gap vs. train loss (log scale).

Table 28: Results for ResNet-18 Trained on CIFAR10 with **batch size 256 and a learning rate of** $1e^{-3}$ while varying the $\rho$ hyperparameter (0.5,0.25,0.05,0.025,0.005,0.0025). Numbers in bold indicate best scores for metrics. For sharpness metrics lower values represent flatter models.

| $\rho$ Value | Generalisation Gap | Test Accuracy | Test ECE | Corruption Accuracy | Prediction Disagreement | Fisher Rao Norm | SAM Sharpness | Relative Flatness |
|---|---|---|---|---|---|---|---|---|
| 0.0000 | 28.050 (0.175) | 0.720 (0.002) | 0.186 (0.001) | 58.614 (0.201) | 0.282 (0.001) | 0.032 (0.001) | 1.366E-05 (1.206E-06) | 34.607 (0.757) |
| 0.5000 | **1.605 (0.646)** | 0.629 (0.024) | 0.079 (0.007) | 52.847 (1.667) | 0.221 (0.009) | 14.767 (0.375) | 6.814E-02 (7.326E-03) | 4156.344 (279.557) |
| 0.2500 | 9.751 (0.640) | **0.835 (0.002)** | **0.026 (0.003)** | **68.479 (0.302)** | **0.089 (0.002)** | 8.712 (0.623) | 3.884E-02 (4.981E-03) | 4876.348 (314.164) |
| 0.0500 | 20.588 (0.125) | 0.794 (0.001) | 0.108 (0.001) | 66.342 (0.164) | 0.168 (0.000) | 0.107 (0.006) | 5.823E-05 (9.056E-06) | 75.093 (1.693) |
| 0.0250 | 22.602 (0.109) | 0.774 (0.001) | 0.124 (0.001) | 64.224 (0.154) | 0.195 (0.000) | 0.065 (0.001) | 2.587E-05 (1.987E-06) | 70.223 (0.941) |
| 0.0050 | 25.793 (0.137) | 0.742 (0.001) | 0.167 (0.001) | 60.985 (0.280) | 0.250 (0.000) | 0.082 (0.009) | 4.861E-05 (7.166E-06) | 57.886 (5.223) |
| 0.0025 | 26.654 (0.130) | 0.733 (0.001) | 0.176 (0.001) | 60.107 (0.226) | 0.262 (0.001) | 0.023 (0.001) | 8.624E-06 (7.512E-07) | 22.262 (0.969) |

## I  EXPLICITLY INCREASING FUNCTION COMPLEXITY

To demonstrate the relationship between sharp minima and increased function complexity, which is suggested by the toy setting results in 4, we employ an experiment in the classification setting on high-dimensional data wherein we artificially increase the complexity of a learning task and record the sharpness of the minima at the end of training. **It is important to note that we calculate the sharpness of the model at the end of training on the same unaltered training dataset for all models**. To conduct this experiment, we randomise the labels in the training dataset of CIFAR 10 in 20% intervals and show that the resulting model trained to minimise loss on this dataset has a sharper minima compared to learning on standard training data (0% randomised data). The training set-up matches that of the baseline models for the ResNet18 with a batch size of 256 and a learning rate of 0.001; however, these results are averaged over 5 seeds (0-4) for each condition.

The increased complexity of the learning task is also reflected in the train loss that is higher for randomised data, despite all models being provided the same training setup, as seen in Table 29. It is important to note that all models achieve 100% train accuracy.

Table 29: Results for ResNet-18 Trained on CIFAR10 with **increasingly randomised data**. For sharpness metrics lower values represent flatter models.

| Percentage of Random Data | Train Accuracy | Train Loss | Fisher Rao Norm | SAM Sharpness | Relative Flatness |
|---|---|---|---|---|---|
| 0% (Baseline) | 1.000 ± 0.000 | 9.997E-05 ±1.381E-06 | 0.031 ±0.001 | 1.344E-05 ±1.968E-06 | 33.700 ±0.949 |
| 20% | 1.000 ±0.000 | 1.039E-04 ±1.714E-06 | 107.830 ±0.183 | 2.594E-01 ±2.349E-03 | 38.615 ±0.693 |
| 40% | 1.000 ±0.000 | 1.071E-04 ±1.759E-06 | 160.742 ±0.292 | 4.180E-01 ±2.405E-02 | 42.111 ±0.613 |
| 60% | 1.000 ±0.000 | 1.085E-04 ±2.294E-06 | 203.816 ±0.212 | 8.927E-01 ±1.848E-02 | 44.990 ±0.972 |
| 80% | 1.000 ±0.000 | 1.107E-04 ±2.535E-06 | 242.238 ±0.260 | 4.984E-01 ±3.913E-02 | 47.011 ±0.878 |
| 100% | 1.000 ±0.000 | 1.094E-04 ±1.490E-06 | 283.181 ±0.129 | 3.450E-01 ±3.319E-02 | 47.095 ±0.543 |

The results from Table 29 show that as the percentage of examples becomes increasingly disjointed through increased randomisation, there is a linear increase in the sharpness of the minima (by Fisher Rao norm and Relative Flatness) found at the end of training. Here, the results simulate a function becoming more complex as the decision boundaries for a particular class become tighter. While in practice this learned function may not be useful, it strongly suggests that the more complex the relationships in the training data, the sharper the minima at the end of training, directly aligning function complexity and geometric properties of minima.

