# OpenReview forum: "A Function Centric Perspective on Flat and Sharp Minima"
_ICLR.cc/2026/Conference — Submitted to ICLR 2026_

### Official Review · Reviewer_29Eb · 2025-10-30

**Soundness:** 2
**Presentation:** 3
**Contribution:** 3
**Rating:** 4
**Confidence:** 4

**Summary:**

The paper investigates the relationship between flatness and generalization from a function-complexity perspective. For that, it empirically analyzes SAM-sharpness, Fisher-Rao-Norm, and Relative Flatness for various networks trained either with standard training, using weight-decay, or data augmentation, as well as using the SAM optimization objective with each of the previous options. The experiments show that when looking at all networks together, there is no correlation between flatness measures and generalization. The paper then argues that flatness should be interpreted as a function-dependent property, not as a universal proxy for generalization.

**Strengths:**

- Investigating how regularization techniques impact flatness measures is sound and interesting.
- The fact that these regularization techniques increase sharpness is insightful.
- Since flatness remains one of the main candidates for explaining why deep networks generalize, this empirical study makes a meaningful contribution to an ongoing and relevant discussion in the field.

**Weaknesses:**

- The paper argues that sharper minima correspond to more complex functions, but this is not formally proven. The example given in section 4 is a good illustration, but it is unclear how it relates to deep learning.
- It is unclear whether flatness metrics computed with and without data augmentation are comparable. Changing the dataset on which flatness is measured changes the entire loss surface. Similarly, using weight decay affects weight norms, which changes the flatness measures, since both Fisher-Rao-Norm and Relative Flatness are norm-based measures.
- Figure 1 (c) could be an instance of Simpson's paradox: while there is a negative correlation over the entire population, there are positive correlations for individual stratums. That is, the correlation inverses as soon as you condition on the training method. Following Judea Pearl's logic, the training method cannot be a mediator, since it is not influenced by flatness. Instead, it must be a confounder, and thus we need to stratify on it. While this could still suggest that regularization causes both flatness and generalization in each stratum, and therefore flatness is _not_ a cause of generalization, as the paper suggests. This needs to be properly tested, though.
- Petzka, et al. 2021 [3] is not an empirical study (line 34), it provides a theoretical explanation about why and when flatness can explain generalization.
- Andriushenko, et al. [2], and Wen et al [5] already established that SAM does not produce flat solutions (line 416).
- The counter-example in Sec. 4 does not work for relative flatness [Petzka]: It requires locally constant labels and this function is not locally constant. Therefore it is not surprising that flatness does not correlate with generalization, here.

Overall, I see this as a flawed but thought-provoking paper that opens the right questions about flatness and generalization but does not yet answer them rigorously.

**Questions:**

- For the results in Tab. 3, do you use the sum loss or mean loss? The large value of relative flatness and Fisher-Rao norm for data augmentation are fairly unusual and could be explained by using the sum loss. In that case, the Hessian over the dataset is the sum of Hessians of each individual sample. Then, data augmentation clearly increases the scale of the value.
- Apart from SAM, weight decay, and data augmentation (line 62), one could also regularize to obtain flatness directly (e.g., [1]). Is there a reason this was not used?
- The relationship between relative flatness and adversarial robustness was recently analyzed by Walter et al. [4]. How do these results relate to this functional view on robustness?
- Walter et al. [4] also uncovered the relation between network confidence and relative flatness. Since minimizing the CE loss leads to high confidence on training examples, this would directly explain Fig. 1 (b).
- Table 1 caption: it should probably read "4 _local_ minima", right?

[1] Adilova, Linara, et al. "FAM: Relative Flatness Aware Minimization." Topological, Algebraic and Geometric Learning Workshops 2023. PMLR, 2023.

[2] Andriushchenko, Maksym, and Nicolas Flammarion. "Towards understanding sharpness-aware minimization." International conference on machine learning. PMLR, 2022.

[3] Petzka, Henning, et al. "Relative flatness and generalization." Advances in neural information processing systems 34 (2021): 18420-18432.

[4] Walter, Nils Philipp, et al. "When Flatness Does (Not) Guarantee Adversarial Robustness." arXiv preprint arXiv:2510.14231 (2025).

[5] Wen, Kaiyue, Tengyu Ma, and Zhiyuan Li. "How Does Sharpness-Aware Minimization Minimizes Sharpness?." OPT 2022: Optimization for Machine Learning (NeurIPS 2022 Workshop).

---

> ### Author Response · Authors · 2025-11-18
> **Response 1**
>
> **W1. The paper argues that sharper minima correspond to more complex functions, but this is not formally proven. The example given in section 4 is a good illustration, but it is unclear how it relates to deep learning.**
>
> Thank you for this question. There is precedence for using toy datasets to understand geometric properties in neural networks that extend to higher complexity problems. [1] explored sharpness in very low complexity datasets such as Swiss Roll and then scaled to more complex tasks to try and fundamentally understand the role of flatness and generalisation. Here, we follow a similar setup to empirically explore the relationship between regulariaation, function complexity and geometric properties of a neural network.
>
> Under Occam’s razor it is stated that a model with fewer assumptions is viewed to be less complex and thus can be described with less precision. This is an argument provided under Minimum Descriptive Length (MDL) from some of the earliest work on Flat Minima by Hochreiter  and Schmidhuber [2] which states that flat minima can be described with less precision than sharp minima.
>
> As a result, a sharp minima under Occam’s razor and MDL are more complex and have a longer MDL than flat minima. That being said, a sharp minima with a longer MDL does not have to generalise better.
>
> Our results in Section 4 illustrate how different functions have different solution space geometry and allow us to better understand our observation in this work  that regaularisation does, indeed, lead to sharper minima over the baseline and performs better across accuracy and safety relevant baselines.
>
> The argument this work, enabled by experiments in Section 4, puts forward is that neither sharpness or flatness is preferential and that  there is no causal relationship between sharpness and generalisation and instead is a property of the function being learned, as observed for the single objective optimisation problems. These insights then hold in the high-dimensional experiments we run. Therefore, we should examine the learned function of the model instead.

---

> ### Author Response · Authors · 2025-11-18
> **Response 2**
>
> **W2. It is unclear whether flatness metrics computed with and without data augmentation are comparable.**
>
> **During our experimentation we also considered this factor please refer to Appendix Section E.1 Augmented or Standard Training Data Sharpness Calculation (ResNet) in the original submission  (ResNet) in the original submission where we show that calculating sharpness metrics Fisher Rao norm, Sam Sharpness and Relative Flatness based on the augmented training data or the standard data causes no statistically significant or substantial difference in the trends observed of higher sharpness recorded for the Augmentation and Augmentation + SAM conditions against the other controls.**
>
> Additionally, we visualise the loss landscapes with augmented training data and non augmented training data for the ResNet and the VGG **(Appendix Section E.1 & F.2)** using augmented training data and normal training data and reaffirm that the visualisations show no core qualitative difference against using augmentation or standard data to visualise the loss landscape. As this paper focuses on observing the trends of increased sharpness when applying regularisation it follows that using the augmented data or the normal data to calculate sharpness does not impact the outcome of our findings.
>
> To further validate what was observed for ResNet across Risher Rao norm, SAM Sharpness and Relative Flatness we have calculated sharpness for each of the ten VGG models trained on CIFAR10 in the Augmentation and Augmentation +SAM using both augmented data and standard data to calculate sharpness.
>
> While there is a slight difference in the core sharpness values, the models in the Augmentation and the Augmentation + SAM condition remain statistically significantly and substantially sharper than all the other baselines, appendix Table 15, regardless of calculating sharpness with the standard training data or augmented data.
>
> The table below shows the sharpness calculations for each of the ten VGG models trained on CIFAR10 in the respective conditions using the standard training data.
>
> | **Condition**      | **Fisher Rao Norm** | **SAM Sharpness**     | **Relative Flatness** |
> |--------------------|---------------------|-----------------------|-----------------------|
> | Augmentation       | 2.322 (0.125)       | 9.589E-03 (1.438E-03) | 481.312 (23.358)      |
> | Augmentation + SAM | 3.756 (0.030)       | 1.497E-02 (7.363E-04) | 1413.712 (22.212)     |
>
> The table below shows the sharpness calculations for each of the ten models in the respective conditions using the augmented training data.
>
> | **Condition** | **Fisher Rao Norm** | **SAM Sharpness**     | **Relative Flatness** |
> |---------------|---------------------|-----------------------|-----------------------|
> | Aug           | 3.505 (0.155)       | 1.967E-01 (2.298E-02) | 688.897 (26.348)      |
> | Aug + SAM     | 4.278 (0.027)       | 9.777E-02 (1.126E-02) | 1609.212 (22.719)     |
>
> We thank you for highlighting this and providing a further opportunity to show that our findings of substantially and statistically significantly sharper models caused under the application of Augmentation and Augmentation + SAM compared to other controls hold regardless of the dataset used for sharpness calculation.
>
> Furthermore, we argue that any regularisation may impact the norms of the model throughout training and in fact this may be the desired effect if we see performance improvements. However, this does not negate the use of Relative Flatness and Fisher Rao norm.
>
>
> **W.3 Figure 1(c) could be an instance of Simpson's paradox**
>
> **As stated in L466 we state explicitly that “Importantly, this highlights the risk of misleading conclusions when aggregating sharpness trends across heterogeneous architectures: we observe that general trends can invert under such aggregation, consistent with Simpson’s Paradox”**, the Figure 1c is supposed to highlight this as for each regulariser we see different trends across seeds, we argue that this means that we should have no general preference towards any particular geometry trend but a focus on what a particular training intervention can do to the sharpness of an architecture from the baseline, which support our function centric perspective that we discuss in the paper and fundamentally motivates our exploration of this.
>
> We will add clarity to our description of Figure 1c to highlight this, showing that flatness and sharpness, depending on the specific control condition, can result in better performance and caution against any particular conclusions related preference of these minima.

---

> ### Author Response · Authors · 2025-11-18
> **Response 3**
>
> **W.4 Figure 1(c) Petzka, et al. 2021 [3] is not an empirical study**
>
> Petzka et al. 2021 is a paper that contains both empirical and theoretical evidence for their proposed relative flatness paper, their empirical evidence showing the robustness of both Relative Flatness and Fisher Rao norm to reparameterisation when all other metrics lose their correlation is particularly compelling. While it is not wholly empirical, its empirical findings presented are core to grounding its theoretical insights and provide the basis of which we employ them in this study.
>
> We will clarify that it is both theoretical and empirical in the paper to improve clarity.
>
> **W.5 Andriushenko, et al. [2], and Wen et al [5] already established that SAM does not produce flat solutions**
>
> We do not claim to be the first to observe this result; however, we do discuss how our findings are counterintuitive to the literature, are counterintuitive to the literature, which continues to suggest that SAM is effective because it finds flatter solutions [3,4,5,6,7].
>
> Furthermore **please see response to 2Sqz Q3 where we explore the impact of the Rho hyperparameter and find that SAM can find flatter minima than the baseline for very small values of rho** but that the best performing model is orders of magnitude sharper than the baseline model. These results are novel and somewhat resolve the tension in literature surrounding SAM ability to find flat and sharp minim and will be included in the paper.
>
> **W.6 The counter-example in Sec. 4 does not work for relative flatness [Petzka]**
>
> The authors understand that Relative Flatness requires locally constant labels. We do not employ this metric for this section. Instead we employ the mean sam sharpness to measure the sharpness of the regression models. **We only employ the measure relative flatness for our classification results in Section 5, in line with locally constant labels required for this measure.**
>
> Section 4 is purely illustrative of the fact that functions have different geometric properties around their global minima, and this changes the final sharpness at the end of training when a neural network fits a specific function. These insights motivate the function centric view that is carried through the paper and used to contextualise the results we observe for the classification tasks, where we observe increased sharpness and utilise the reparameterisation invariant metrics of Relative Flatness and Fisher Rao norm.

---

> ### Author Response · Authors · 2025-11-18
> **Response 4**
>
> **W.7 Overall, I see this as a flawed but thought-provoking paper that opens the right questions about flatness and generalization but does not yet answer them rigorously.**
>
> Thank you for noting that our work is thought-provoking and asks the right questions.
>
> However, could you please elaborate on how our work lacks empirical rigor?
>
> We display empirical rigour in the following ways:
>
> 1. We provide an experimental setup that first isolates the impact of several single-objective optimisation functions on a neural network within a fixed training budget and initialisation. Allowing us to understand how data complexity itself governs the sharpness of the resulting model throughout training. We train 1400 models to conduct this evaluation of training in a fixed training budget and specific training losses.
> 2. We perform multiple independent studies of how independently applied regularisation methods impact the geometry found at the end of training. We do this to remove confounding factors that would invalidate our findings.
> 3. We explore three vision datasets (CIFAR10, CIFAR100, and TinyImageNet) to observe how our findings scale under increased task complexity.
> 4. For the high-dimensional learning tasks, CIFAR10, CIFAR100, and TinyImageNet, we explore across the 3 architectures and train 10 models (on seeds 0-9) for each architecture  to have a large sample size to make our findings robust. Overall, on high-dimensional tasks, we trained 480 models to ensure our results were robust, incurring significant computational costs to ensure empirical rigour.
> 5. Furthermore, to ensure that our results were not artifacts of special training hyperparameters of learning rate and batch size, we perform a hyperparameter sweep for the Resnet and VGG19 architectures. To this end, we explore the learning rates of 0.01 and 0.001 as well as the batch sizes 128 and 256 to ensure that our findings are general and robust. This process led to the further training of 360 models to verify these findings. This results in a total number of 840 for the high-dimensional datasets in this study.
> 6. We also ensure that we report the Standard Error of the Means; it is more reflective of the error of the mean, which is more informative than the dispersion of the data presented by Standard Deviation [8], as it provides a more accurate representation of the mean.
> 7. In our work, we also explore 3 different metrics (robustness, calibration, and functional diversity) to evaluate models beyond traditional accuracy and loss to ensure that we provide the fullest empirical assessment of the models.
> 8. We also verify sharpness metrics across a range of metrics (Relative Flatness, Fisher Rao norm, and Sam Sharpness) and visualisations to ensure that trends remain stable, as opposed to only checking a singular metric, which could provide misleading insights.
> 9. We use reparametrisation metrics to ensure that our findings are robust.
> 10. Finally, we validate the calculation of sharpness metrics on both augmented and standard training data to ensure that our results are representative of true sharpness values that would be expected for the augmentation condition.
>
> **As far as we are aware, our study represents the most comprehensive controlled empirical examination of geometric properties of neural networks and their relation to sharpness via regularisation controls across dataset complexities, different inductive biases, and hyperparameter settings.**
>
> What further rigour do you require, and can you highlight specific literature that exceeds our setup?
>
> **Q1 For the results in Tab. 3, do you use the sum loss or mean loss?**
>
> We use the mean loss over the sum of the loss for the sharpness calculations.
>
> **For concerns regarding augmentation, please see Appendix Section E.1 for ResNet and Section F.2 for VGG in the original submission.**

---

> ### Author Response · Authors · 2025-11-18
> **Response 5**
>
> **Q2 Is there a reason FAM was not used?**
>
> The core aim of this work was to show that assumptions relating to the causality of flatness and generalisation are flawed.
>
> Since the initial reparameterisation invariance literature that showed sharp minima can generalise [9], new metrics (Relative Flatness [10] and Fisher Rao norm [11]) have reaffirmed the relationship between generalisation and flatness. Through analysing the use of commonly used regularisation techniques (e.g., SAM, weight decay, augmentation), we observe in a controlled setting that when these regularisers improve generalsiation they also induce sharper minima. Concretely, this rejects the notion that flatness is a prerequisite for a well-performing model.
>
> FAM is not a commonly used regulariser; according to Google Scholar, it only has 3 citations compared to Augmentation [12] with 3672, weight decay [13] with 34044, and SAM [14] with 2065. In this paper, we analyse how commonly used methods of regularisation impact generalisation, safety, and landscape geometry across datasets and architectures. Furthermore, including another condition would vastly increase the number of models that require training and analyzing.
>
> Finally, the use of FAM, which operates to reduce relative flatness directly, would reduce the rigor of this research as it would bias the results towards a flatter result for a specific metric that is being evaluated as part of the optimisation process and therefore introduces a form of metric hacking.
>
> For these reasons, we exclude FAM from our evaluation.
>
> **Q3 "The relationship between relative flatness and adversarial robustness was recently analyzed by Walter et al. [4]. How do these results relate to this functional view on robustness?"
> &
> Q4 "Walter et al. [4] also uncovered the relation between network confidence and relative flatness. Since minimizing the CE loss leads to high confidence on training examples, this would directly explain Fig. 1 (b)."**
>
> Thank you for highlighting the work by Walter et al.
>
> **However, the work conducted by Walter et al is contemporaneous by the definition of the ICLR review process, as it was published on the 16th of October on Arxiv after the ICLR deadline of  September 24th, 2025.** Furthermore, **an internet search for the Walter et al. paper shows that it is also currently under peer review at ICLR 2026.**
>
> According to the ICLR Reviewer Guidelines, comparison to this study is not required, and our work must not be evaluated against it.
>
> **Could you please clarify your questions based on this information and assess the novelty of our findings according to the ICLR review process?**
>
>
> **Q5 Table 1 caption: it should probably read "4 local minima", right?**
>
> The caption on this table is correct, the Himmelblau’s function has four minima, each of which represents a global minima for the function. Other literature studying this function have described the Himmelblau’s function to have four global minima [15].

---

> ### Author Response · Authors · 2025-11-18
> **Response 6 (Final)**
>
> **References:**
>
> [1] Huang, W.R., Emam, Z., Goldblum, M., Fowl, L., Terry, J.K., Huang, F. and Goldstein, T., 2020. Understanding generalization through visualizations.
>
> [2] Hochreiter, S. and Schmidhuber, J., 1997. Flat minima. Neural computation, 9(1), pp.1-42.
>
>
> [3] Ramasubramanian, S., Freed, B., Capone, A. and Schneider, J., Improving Model-Based Reinforcement Learning by Converging to Flatter Minima. In The Thirty-ninth Annual Conference on Neural Information Processing Systems.
>
> [4] Kim, M., Li, D., Hu, S.X. and Hospedales, T., 2022, June. Fisher sam: Information geometry and sharpness aware minimisation. In International Conference on Machine Learning (pp. 11148-11161). PMLR.
>
> [5] Seowon, J., Seunghyun, M., Jiyoon, S. and Sangwoo, H., 2025. Achieving Distributional Robustness with Group-Wise Flat Minima. Mathematics, 13(20), p.3343.
>
> [6] Mi, P., Shen, L., Ren, T., Zhou, Y., Sun, X., Ji, R. and Tao, D., 2022. Make sharpness-aware minimization stronger: A sparsified perturbation approach. Advances in Neural Information Processing Systems, 35, pp.30950-30962.
>
> [7] Long, P.M. and Bartlett, P.L., 2024. Sharpness-aware minimization and the edge of stability. Journal of Machine Learning Research, 25(179), pp.1-20.
>
> [8] Belia, S., Fidler, F., Williams, J. and Cumming, G., 2005. Researchers misunderstand confidence intervals and standard error bars. Psychological methods, 10(4), p.389.
>
> [9] Dinh, L., Pascanu, R., Bengio, S. and Bengio, Y., 2017, July. Sharp minima can generalize for deep nets. In the International Conference on Machine Learning (pp. 1019-1028). PMLR.
>
> [10] Petzka, H., Kamp, M., Adilova, L., Sminchisescu, C. and Boley, M., 2020. Relative Flatness and Generalization. arXiv preprint arXiv:2001.00939.
>
> [11] Liang, T., Poggio, T., Rakhlin, A. and Stokes, J., 2019, April. Fisher-rao metric, geometry, and complexity of neural networks. In The 22nd international conference on artificial intelligence and statistics (pp. 888-896). PMLR.
>
> [12] Perez, L. and Wang, J., 2017. The effectiveness of data augmentation in image classification using deep learning. arXiv preprint arXiv:1712.04621.
>
> [13] Loshchilov, I. and Hutter, F., 2017. Decoupled weight decay regularization. arXiv preprint arXiv:1711.05101.
>
> [14] Foret, P., Kleiner, A., Mobahi, H. and Neyshabur, B., 2020. Sharpness-aware minimization for efficiently improving generalization. arXiv preprint arXiv:2010.01412.
>
> [15] Cerino, F., Diaz-Pace, J.A., Tassone, E.A., Tiglio, M. and Villegas, A., 2023. Hyperparameter Optimization of an hp-Greedy Reduced Basis for Gravitational Wave Surrogates. Universe, 10(1), p.6.

---

### Official Review · Reviewer_2jxP · 2025-10-31

**Soundness:** 2
**Presentation:** 3
**Contribution:** 3
**Rating:** 6
**Confidence:** 5

**Summary:**

The paper offers an alternative perspective on the flatness of the loss surface, a property long regarded as an indicator of good generalization. Specifically, it shows that the flatness of the obtained solution decreases when the learned function becomes simpler, and conversely, that more complex functions correspond to sharper minima. The authors argue that regularization techniques that improve model performance tend to produce sharper solutions, thereby providing empirical evidence that models representing more complex functions exhibit greater sharpness across different levels of regularization.

**Strengths:**

The paper addresses the entanglement between flatness\sharpness of loss surface and generalization of neural networks from an interesting perspective of the function complexity. The empirical results advocate for considering that more complex functions learned by networks can result in sharper solutions, without declining generalization, meaning that flatness has to be considered depending on the hardness of the task.

**Weaknesses:**

I find the first part of the empirical experiments somewhat misleading. In lines 54–58, these experiments are introduced as showing that the complexity of the objective determines how sharp the resulting solution will be. However, it is important to distinguish that, in this group of experiments, the known functions define the target function that the network is supposed to learn but not the objective (loss) function being optimized. In other words, the optimization does not occur directly on the surface of these proposed functions but rather on the loss surface defined by the mean squared error (MSE) in a regression setting. This distinction is not clearly conveyed, and the same confusion appears in the description of the experiments in Section 4.

Furthermore, one of the key contributions of [1] is the assumption of locally constant labels, which must hold for flatness to be a meaningful measure of generalization. This assumption is reasonable in classification tasks, but not necessarily in regression tasks, especially when the target functions exhibit high curvature. Consequently, this assumption breaks down in the first group of experiments using these functions.

In the introduction, the experiments are described as being designed to evaluate adversarial robustness, yet the presented empirical results concern natural noise robustness. This inconsistency should be clarified.

While I agree that sharpness measurements should not be taken as an absolute indicator of generalization, I disagree with the claim that this can be demonstrated by comparing sharpness values across different tasks or setups. For instance, data augmentation substantially alters the geometry of the loss landscape, making direct comparison of sharpness between augmented and non-augmented training runs questionable. Within a single setup, sharper solutions might indeed generalize worse, but comparing sharpness across heterogeneous settings risks introducing misleading effects, like Simpson’s paradox, as the paper itself cautions. This issue also makes the results involving SAM regularization appear contradictory: the experiments show higher SAM-sharpness even when SAM is explicitly designed to regularize and reduce it.

[1] Petzka, Henning, et al. “Relative Flatness and Generalization.” NeurIPS 34 (2021): 18420–18432.

**Questions:**

1 - In classical machine learning regularization is supposed to restrict the class of learned functions, usually to a simpler functions subset (like in case of L2 regularization for example). Your work claims that regularization increases complexity of the learned functions. Can you please elaborate on this contradiction?

2 - Your main contribution is stated as demonstrating that geometry of minima reflects complexity of the learned function, but you do not have any definition of the complexity - can you propose any?

3 - In the captures to the visualizations you have a mention of "51 random directions". What do you mean by it?

---

> ### Author Response · Authors · 2025-11-18
> **Response 1**
>
> **W1. Section 4 clarity**
>
> Thank you for highlighting this. We will make this distinction clearer in the camera-ready version. We were trying to convey the following:
>
> Figures 2 and 3 and tables 1 and 2 demonstrate that a neural network (or any model)  that would consist of 2 parameters could be optimised on these landscapes to find global minima. In the case of Himmelblau’s function, no particular global minima is preferred over another. Even though each minima has different characteristics, i.e, sharper/flatter, we then use Figures 3 and Table 2 to highlight that different functions' global minima have different properties, but are all equally global minima.
>
> Then the results when learning the surface using mean squared error in the regression setting are showing that how the surface of the function that is being learned can affect the sharpness that the model achieves when provided the same training setup and budget, which results in similar/the same generalisation gaps being achieved, however, with differing sharpness of the minima found.
>
>  **W2.  The assumption of locally constant labels for Section 4**
>
> Thank you for highlighting this. For section 4, we do not use the metric Relative Flatness to evaluate the minima found by the model for this exact reason. As a result, we use the metric SAM Sharpness, which provides an approximate understanding of the local sharpness of the model at its minima for each function. We take this understanding from our results in Section 4 to hypothesize how regularisation may impact sharpness for classification tasks, CIFAR10,CIFAR100, and TinyImageNet, where we do use the metric Relative Flatness.
>
> Concretely, the function centric lens provided by Section 4 on the regression tasks where we can visualize the global minima provide a strong insight into how fitting a function can change the local geometry of minima that a neural network finds, we find that this analysis and understanding can be carried into our high-dimensional experiments where we do employ Relative Flatness and Fisher Rao norm, reparameterisation invariant sharpness metrics.
>
>  **W3.  Adversarial robustness, yet the presented empirical results concern natural noise robustness.**
>
> Thank you for highlighting this. We do acknowledge that our evaluation using CIFAR10-C and CIFAR100-C does not consider worst-case adversarial perturbation and focuses more on average-case performance on corruptions. However, some of the corruptions, especially at the highest severity category of 5, cannot be considered purely natural as they can largely obfuscate the original sample. We will update the description such that it better reflects the natural noise robustness you have correctly referenced.
>
>  **W4.1. For instance, data augmentation substantially alters the geometry of the loss landscape, making direct comparison of sharpness between augmented and non-augmented training runs questionable.**
>
> We thank the reviewer for understanding our perspective that sharpness measurements should not be taken as an indicator of generalisation.
>
> During our experimentation we also considered this factor **please refer to Appendix Section E.1 Augmented or Standard Training Data Sharpness Calculation (ResNet) in the original submission** where we show that calculating sharpness metrics Fisher Rao norm, Sam Sharpness and Relative Flatness based on the augmented training data or the standard data causes no statistically significant or substantial difference in the trends observed of higher sharpness recorded for the Augmentation and Augmentation + SAM conditions against the other controls.
>
>  **W4.2. Comparing sharpness across heterogeneous settings risks introducing misleading effects, like Simpson’s paradox**
>
> As stated in L466 we state explicitly that “Importantly, this highlights the risk of misleading conclusions when aggregating sharpness trends across heterogeneous architectures: we observe that general trends can invert under such aggregation, consistent with Simpson’s Paradox”, the Figure 1c is supposed to highlight this as for each regulariser we see different trends across seeds, we argue that this means that we should have no general preference towards any particular geometry trend but a focus on what a particular training intervention can do to the sharpness of an architecture from the baseline, which support our function centric perspective that we discuss in the paper and fundamentally motivates our exploration of this.
>
> We will add clarity to our description of Figure 1c to highlight this, showing that flatness and sharpness, depending on the specific control condition, can result in better performance and caution against any particular conclusions related preference of these minima.

---

> ### Author Response · Authors · 2025-11-18
> **Response 2**
>
> **W4.3. SAM regularization appear contradictory**
>
> SAM has been shown to reduce sharpness using non reparameterisation metrics, however in our study we use reprameterisation invariant metrics to study its impacts of the loss surface. **This is fully elaborated on in our response to reviewer 2Sqz under Q3**. Furthermore, our metric of SAM sharpness is wholly different from what SAM optimises for, in-line with our definition, SAM Sharpness is the average case sharpness of random points rho away from the minima, whereas SAM optimisation is gradient calculations from worst case adversarial weight spaces applied to the parameters of the model during training. In addition it does not directly optimise to reduce the SAM Sharpness metric that we use in this paper.  As a result, our results add further nuance to understanding SAM beyond the assumption that it minimises sharpness.
>
> **Q1. In classical machine learning regularization is supposed to restrict the class of learned functions, usually to a simpler functions subset (like in case of L2 regularization for example). Your work claims that regularization increases complexity of the learned functions. Can you please elaborate on this contradiction?**
>
> Under Occam’s razor it is stated that a model with fewer assumptions is viewed to be less complex and thus can be described with less precision. This is an argument provided under Minimum Descriptive Length (MDL) from some of the earliest work on Flat Minima by Hochreiter  and Schmidhuber [1] which states that flat minima can be described with less precision than sharp minima.
>
> As a result a sharp minima, under occams razor and MDL are more complex and have a longer MDL than flat minima. What we observe in our experiments is that regaularisation does, indeed, lead to sharper minima over the baseline and performs better across accuracy and safety relevant baselines.
> Our takeaway from this is regularisation can constrain a model to reach a sharper minima which is considered more complex with a longer minimum descriptive length than a flat minima. Therefore, our findings are not in contention but in agreement with classical machine learning literature as constricting an overparameterised model can increase complexity of the learned function while constrain its representation space to be more complex under our functional perspective.
>
> That being said a sharp minima with a longer MDL does not have to generalise better - but we do observe that often it does. The argument this work puts forward is that neither sharpness or flatness is preferential and that there is no causal relationship between sharpness and generalisation. Therefore, we should examine the learned function of the model instead.

---

> ### Author Response · Authors · 2025-11-18
> **Response 3**
>
> **Q2. Your main contribution is stated as demonstrating that geometry of minima reflects complexity of the learned function, but you do not have any definition of the complexity - can you propose any?**
>
> The statement regarding complexity being reflected in the learned function relates to the fundamental understanding of flat and sharp minima offered by Occam's Razor.
>
> Occam's Razor states that of two competing hypotheses, the one that makes the fewest assumptions is preferred. With regard to neural network minima, it is understood that flatter minima are preferential due to the fact that they can be provided with less precision and therefore have less reliance on assumptions. This was first posited by [1] that related flatness of minima to a shorter Minimum Descriptive Length; this notion has been largely reinforced in literature since, and flat minima have been considered preferential.
>
> As a result, the antipodal sharp minima are considered to have more assumptions and thus require a higher degree of precision to describe their function, in line with the fundamental arguments provided by [1]. Therefore, through this understanding, sharp minima can be considered as representing a more complex function.
>
> Measuring such qualities through the lens of Occam’s razor means there are no precise metrics. However, measuring the geometric properties of the minima at the end of training has been provided by literature as a way to interpret this; as a result, we use the reparameterisation invariant sharpness metrics to do this in our paper.
>
> Function complexity with respect to a training dataset can be considered from many perspectives, such as the number of classes and how many of the data points contain shared features. For example, Huang et al [2] show that on the toy Swiss Roll dataset that models learning on random data fit tighter decision boundaries for a 6-layer MLP; they use this to argue that sharp minima relate to memorisation. However, under our function centric view, it simply relates to a more complex function being learned.
>
> Our function centric view argues that the model learning the memorised solution happens to be more complex due to the lack of alignment between examples, which would be expected as class labels are sampled uniformly over examples with disjoint features.
>
> To this effect we have added an experiment where we randomise the examples in the trainset of CIFAR 10 in 20% intervals and show that the resulting model has a sharper minima compared to learning on unrandomised data points (0% randomised data), this is also reflected in the train loss that is higher for randomised data despite all models being provided the same training setup, yet they all achieve 100% train accuracy.
>
> | **Proportion Random Training Data** | **Train Accuracy** | **Train Loss**        | **Fisher Rao Norm** | **SAM Sharpness**     | **Relative Flatness** |
> |-------------------------------------|--------------------|-----------------------|---------------------|-----------------------|-----------------------|
> | 0% (Baseline)                       | 1.000 (0.000)      | 9.997E-05 (1.381E-06) | 0.031 (0.001)       | 1.344E-05 (1.968E-06) | 33.700 (0.949)        |
> | 20%                                 | 1.000 (0.000)      | 1.039E-04 (1.714E-06) | 107.830 (0.183)     | 2.594E-01 (2.349E-03) | 38.615 (0.693)        |
> | 40%                                 | 1.000 (0.000)      | 1.071E-04 (1.759E-06) | 160.742 (0.292)     | 4.180E-01 (2.405E-02) | 42.111 (0.613)        |
> | 60%                                 | 1.000 (0.000)      | 1.085E-04 (2.294E-06) | 203.816 (0.212)     | 8.927E-01 (1.848E-02) | 44.990 (0.972)        |
> | 80%                                 | 1.000 (0.000)      | 1.107E-04 (2.535E-06) | 242.238 (0.260)     | 4.984E-01 (3.913E-02) | 47.011 (0.878)        |
> | 100%                                | 1.000 (0.000)      | 1.094E-04 (1.490E-06) | 283.181 (0.129)     | 3.450E-01 (3.319E-02) | 47.095 (0.543)        |
>
> The results show that the model becomes sharper linearly (by Fisher Rao norm and Relative Flatness) as it learns on increased randomised data, i.e., the function becomes more complex and thus the decision boundaries are tighter and the model is sharper.
>
> Finally, it is important to note that other sources of increased complexity can include an increased number of classes or a higher variety of samples in a particular class, which we provide via augmentation, which increases the sharpness of the minima found at the end of training over the baseline condition, as shown in our experiments.
>
> **We thank you very much for raising this question. It has allowed us to further expand on our argument and provide stronger evidence for the function-centric position of this paper. The new results will be included in the main body of the paper.**

---

> ### Author Response · Authors · 2025-11-18
> **Response 4 (Final)**
>
> **Q3. In the captures to the visualizations you have a mention of "51 random directions". What do you mean by it?**
>
> To clarify what we meant by this statement, we perturb the model in 2 random directions that are approximately orthogonal, filter-wise normalised directions in the weight space, as done by Li et al, 2018 [3]. We then explore the loss in the domain of the perturbations [-1,1]^2 with 51 steps in both directions. We will update the caption to reflect this, thank you for allowing us to improve the clarity of these figures.
>
> **References**
>
> [1] Hochreiter, S. and Schmidhuber, J., 1997. Flat minima. Neural computation, 9(1), pp.1-42.
>
> [2] Huang, W.R., Emam, Z., Goldblum, M., Fowl, L., Terry, J.K., Huang, F. and Goldstein, T., 2020. Understanding generalization through visualizations.
>
> [3] Li, H., Xu, Z., Taylor, G., Studer, C. and Goldstein, T., 2018. Visualizing the loss landscape of neural nets. Advances in neural information processing systems, 31. https://proceedings.neurips.cc/paper/2018/hash/a41b3bb3e6b050b6c9067c67f663b915-Abstract.html

---

### Official Review · Reviewer_2Sqz · 2025-11-01

**Soundness:** 2
**Presentation:** 2
**Contribution:** 2
**Rating:** 2
**Confidence:** 3

**Summary:**

In this paper, the authors focus on the relationship between model sharpness and performance. They revisit the common belief that flatter minima lead to better generalization and instead propose a function-centric perspective: the geometry of the solution (e.g., sharpness) reflects the complexity of the learned function rather than directly determining performance. The authors conduct both toy experiments and real data experiments (on CIFAR-10, CIFAR-100, and TinyImageNet), comparing the performance of several regularization techniques such as weight decay, data augmentation, and Sharpness-Aware Minimization (SAM).

**Strengths:**

-	Understanding the relationship between sharpness and a model’s performance and complexity is an interesting research question.
-	The paper is well-structured overall and easy to follow.

**Weaknesses:**

-	In several places, the writing and presentation of the paper could be improved. See the questions section below for details.
-	I’m not sure about the motivation of Section 4. The comparisons between different objectives don’t make much sense to me. These problems are quite different from each other, so it’s not surprising that the flatness of their minima varies. This seems inconsistent with the motivation in the context of deep learning optimization, where the training objective is the same (or at least similar) but different algorithms are used to optimize it.
-	The precise meaning of function space complexity is unclear. From the discussion in Section 5, it seems to refer to several evaluation metrics (not only just test accuracy) such as accuracy and calibration error (ECE). If that’s the case, the experiments simply suggest that adding more regularization techniques (e.g., weight decay, data augmentation, SAM) improves performance.
-	See more in the questions section below.

**Questions:**

-	There are two contributions listed from the bottom of page 2 to the top of page 3, and they seem quite similar. Is this a repetition? It might be good to check.
-	Regarding the sharpness metric, I wonder why basic measures such as the largest eigenvalue of the Hessian or the trace of the Hessian are not used. The current metrics are fine, but including the most common ones could make the analysis more complete.
-	Related to the above, the original SAM paper shows that SAM tends to converge to points with smaller largest eigenvalues of the Hessian. This seems to contradict the message conveyed in the current paper. I would appreciate it if the authors could comment on this.
-	Many metrics mentioned in Appendices B and C are not formally defined as stated in the main paper. It would be helpful to include formal definitions rather than only referring to other papers or code, for clarity and completeness.
-	I’m also curious about the setup of the baseline. For example, with ResNet-18, the test accuracy is typically above 90% on CIFAR-10 and above 70% on CIFAR-100.

---

> ### Author Response · Authors · 2025-11-18
> **Response 1**
>
> **W1. Motivation of Section 4.  These problems are quite different from each other, so it’s not surprising that the flatness of their minima varies.**
>
> Thank you for raising this point. You are correct in understanding that these minima have different properties; this is exactly what this section seeks to highlight - that different problems require different geometric solutions.
>
> Section 4 is intended to illustrate, as you have correctly inferred, that different functions have different landscapes around their minima; this is to be expected as different functions have different complexities. We then show that fitting a neural network to these landscapes maps the relationship well. The results show that neural networks generally have a relationship between increased sharpness and the complexity of the function that is being learned. Through this, it is possible to show that the neural network's learning process is related to functional complexity.
>
> As a result, when we then scale our experiments to high-dimensional tasks in computer vision, we take the perspective gained from Section 4 to derive hypotheses, analyse, and understand the results we see relating to the sharpener minima found at the end of training under the application of regularization.
>
> Here, we see a relationship between increased sharpness and model performance. This is not because sharpness is a causal factor for generalisation. Under the application of regularization techniques, the neuron networks' function increases in complexity as they allow the model to traverse the landscape and better map the complex relationship between the data and its label to get improved performance that corresponds to a sharper minima than the baseline model can reach. We will clarify the relationship between section 4 and the results of our computer vision tasks in this paper, such that this is explicit.
>
> Furthermore, there is precedence for this type of exploration, as [1] explored sharpness in very low complexity datasets such as Swiss Roll and then scaled to more complex tasks to try and fundamentally understand the role of flatness and generalisation.
>
> **W2.The precise meaning of function space complexity is unclear.**
>
> We thank the reviewer for allowing us to clarify the distinction between our performance and safety evaluations alongside our complexity assessment conduct through the use of  sharpness metrics.
> Neural networks are universal function approximators, under the universal approximation theorems, as they can map any function (input output output pair). A model's function can be evaluated in several ways, including test accuracy, calibration error, prediction disagreement, and robustness.
>
> The evaluations we conduct regarding robustness, calibration, and prediction disagreement are safety evaluations that provide more insights into how well the model performs given a training set up beyond accuracy, considering questions of real-world deployment.
>
> Currently, literature uses sharpness metrics to assess the complexity of a model. Reparameterisation invariant metrics, such as those we explore in the paper, have been employed to suggest a causal relationship between flatness and model performance (under a range of evaluations).
>
> The mapping of input to output (the model function) can be modified in a variety of ways based on the training data, the model's implicit bias, or any regularisation that is applied during the training process.
>
> We posit, from our initial experiments in Section 4, that changing the training regime of a model based upon regularisation enables improved function approximation. We see that the use of SAM and Augmentation in particular enables improved performance (across evaluations) while reaching sharper minima than a model trained without these regularisers. This goes strongly against the perspective that flat minima are causally related to smaller generalization gaps.
>
> Therefore, our work shows that applying regularisers does improve performance, as expected; however, they cause the model to reach a sharper minima (which is unexpected in current literature).
>
> As a result, we argue that this performance increase is derived through a better input and output mapping of a complex function. This is shown in the sharper minima reached at the end of training compared to the baseline condition.

---

> ### Author Response · Authors · 2025-11-18
> **Response 2**
>
> **Q1. two contributions listed from the bottom of page 2 to the top of page 3, and they seem quite similar.**
>
> The first contribution stated in the paper discussed the findings from single object optimisation problems, which illustrate how different functions have different complexities and that this is reflected in the landscape surrounding their global minima.
>
> The second contribution refers to high-dimensional problems wherein employing regularisers (weight decay, augmentation and SAM) can allow the optimisation process to converge on sharper minima that often exhibit improved accuracy and safety evaluations. Provided that these results cover different task complexities and experimental setups, it is important to highlight how the findings, although similar in outcome, stem from different parts of the paper.
>
> To improve the conciseness and clarity of writing, we will combine both of these contributions while keeping the distinction above clear.
>
> Again, thank you for aiding us to improve the clarity of the paper and the key takeaways.
>
> **Q2.  I wonder why basic measures such as the largest eigenvalue of the Hessian or the trace of the Hessian are not used.**
>
> In our analysis, we mainly focus on sharpness metrics that are reparameterisation invariant. Dinh et al., [2] have shown that Hessian based metrics such as the eigenvalue of the Hessian and the trace of the Hessian, are not reparameterisation invariant measures and that they can be manipulated via linear reparameterisations that do not change the function of the model but can make the minima appear sharper. This therefore, undermines the use of non reparameterisation invariant metrics when assessing the geometric properties of a neural network minima.
>
> Furthermore, Petzka et al., [3] reaffirmed this by showing that the Trace of the Hessian and PAC-Bayes flatness are not reparameterisation invariant, but that Relative Flatness and Fisher Rao norm [4] are. Therefore, we use Relative Flatness and Fisher Rao norm to ensure that our study, and its findings are robust (to reparametrisations).
>
> Petzka et al., [3] reaffirm the relationship between generalisation and flatness with Relative Flatness and Fisher Rao norm as reparameterisation invariant metrics.
>
> Our study primarily focuses on how these reparameterisation invariant metrics are impacted by regularisation and how this in turn, impacts the geometry of the minima that are found at the end of training and what this tells us about generalisation and flatness.

---

> ### Author Response · Authors · 2025-11-18
> **Response 3**
>
> **Q3. Original SAM paper shows that SAM tends to converge to points with smaller largest eigenvalues of the Hessian.**
>
> Thank you for asking this question and allowing us to further comment on the disconnect between the original SAM paper and our findings. This question has directly led to a set of experiments that further demystify the relationship of SAM and sharpness, which can be found below, highlighting that SAM can find flatter minima, but that often the best performing model is that which is sharper when compared to the baseline.
>
> We find that for most models trained with SAM, they converge to sharper minima under the core sharpness metrics that we have utilised in this paper. In the original SAM paper, they do not explore the reparameterisation invariant sharpness metrics that we have explored in our paper. Our motivation for doing this is that [2] showed that non reparametrisation invariant metrics can be artificially modified to be flatter or sharper without impacting the model's function and performance. As a result, our use of such reparametrisation invariant metrics could be the source of the misalignment between our findings and the message conveyed in the SAM paper.
>
> To ensure that our findings with SAM are not an artifact, the rho hyperparameter (which defines the perturbation radius) used in the paper, we have spanned rho values from its default 0.05 to understand how this parameter impacts the sharpness of the resulting model.
>
> The table below shows, averaged across ten models for each condition, the results of performing a sweep on the rho hyperparameter. Please note that the rho of 0.000 represents the baseline model trained without SAM.
>
> | **Rho Parameter Values** | **Generalisation Gap** | **Test Accuracy** | **Test ECE**      | **Corruption Accuracy** | **Prediction Disagreement** | **Fisher Rao Norm** | **SAM Sharpness**     | **Relative Flatness** |
> |--------------------------|------------------------|-------------------|-------------------|-------------------------|-----------------------------|---------------------|-----------------------|-----------------------|
> | 0.000 (Baseline no SAM)  | 28.050 (0.175)         | 0.720 (0.002)     | 0.186 (0.001)     | 58.614 (0.201)          | 0.282 (0.001)               | 0.032 (0.001)       | 1.366E-05 (1.206E-06) | 34.607 (0.757)        |
> | 0.5000                   | **1.605 (0.646)**      | 0.629 (0.024)     | 0.079 (0.007)     | 52.847 (1.667)          | 0.221 (0.009)               | 14.767 (0.375)      | 6.814E-02 (7.326E-03) | 4156.344 (279.557)    |
> | 0.2500                   | 9.751 (0.640)          | **0.835 (0.002)** | **0.026 (0.003)** | **68.479 (0.302)**      | **0.089 (0.002)**           | 8.712 (0.623)       | 3.884E-02 (4.981E-03) | 4876.348 (314.164)    |
> | 0.0500                   | 20.588 (0.125)         | 0.794 (0.001)     | 0.108 (0.001)     | 66.342 (0.164)          | 0.168 (0.000)               | 0.107 (0.006)       | 5.823E-05 (9.056E-06) | 75.093 (1.693)        |
> | 0.0250                   | 22.602 (0.109)         | 0.774 (0.001)     | 0.124 (0.001)     | 64.224 (0.154)          | 0.195 (0.000)               | 0.065 (0.001)       | 2.587E-05 (1.987E-06) | 70.223 (0.941)        |
> | 0.0050                   | 25.793 (0.137)         | 0.742 (0.001)     | 0.167 (0.001)     | 60.985 (0.280)          | 0.250 (0.000)               | 0.082 (0.009)       | 4.861E-05 (7.166E-06) | 57.886 (5.223)        |
> | 0.0025                   | 26.654 (0.130)         | 0.733 (0.001)     | 0.176 (0.001)     | 60.107 (0.226)          | 0.262 (0.001)               | 0.023 (0.001)       | 8.624E-06 (7.512E-07) | 22.262 (0.969)        |
>
> Here, the results show that only the smallest value of rho of 0.0025 leads to a model that  has improved performance and is flatter across all metrics when compared to the baseline.
>
> However, the model that performs best is that of a 0.2500 rho value, which leads to the best accuracy, performs the best on safety evaluations, and is statistically significantly substantially sharper than the baseline model by orders of magnitude.
>
> These results somewhat resolve the tension you have raised, even under the reparameterisation invariant metrics, SAM can lead to a flatter minima than the baseline - it just happens to be the case that the best performing model using SAM is orders of magnitude sharper than the baseline.
> Therefore, SAM can find flatter minima; however, this is dependent on the rho hyperparameter, and to get the best performance, a sharper minima is likely to be found.
>
> **We thank you for asking this question, as it further strengthens our finding that flatness is not causally related to generalisation, these results will be referenced in the main body and included in the appendix of the camera ready to add to our paragraph “SAM Can Find Sharper Minima” in the Section 5.**

---

> ### Author Response · Authors · 2025-11-18
> **Response 4 (Final)**
>
> **Q4. It would be helpful to include formal definitions rather than only referring to other papers or code, for clarity and completeness.**
>
> While we do define the main metrics that we use in the paper, for completeness we will add the mathematical notation for the more widely known metrics  described in appendix section C.  We will also provide the notation for Relative Flatness from [2] in Appendix Section B.
>
> Thank you for highlighting this, we will ensure completeness of all metric definitions to ensure that our paper is self-contained and high-utility research.
>
> **Q5. I’m also curious about the setup of the baseline.**
>
> **As stated in Appendix Section B (L846-L848) for CIFAR10**, “We use SGD with the momentum hyperparameter at 0.9 to minimize cross entropy loss for 100 epochs, using a batch size of 256 and a learning rate of 0.001.” **The settings for CIFAR100 are described in (L864-L866)**, with the only differences being a learning rate of 0.01 and the use of the Cosine Annealing learning rate scheduler. These configurations set the foundation for our evaluation of model behavior.
>
> **Here, both models represent a vanilla setup without regularisers.** This shows the base capacity of the model under these conditions. **As stated in L318-320 in the Section Experimental Setup: “Our objective is to characterise, under controlled conditions, the geometric and safety effects of regularisation controls, not to optimise for state-of-the-art performance.”** This setup has allowed us to gain a nuanced and novel perspective on the relationship between geometric properties of a model and generalisation.
>
> **In Appendix E.2 and F.1, we show that varying batch size and learning rate in ResNet and VGG architectures affects the minima reached by the models. Our findings indicate that regularisation generally improves performance, as expected, but leads to sharper minima than the baseline, contrary to current assumptions about the desirability of flat minima.**
>
> **References:**
>
> [1] Huang, W.R., Emam, Z., Goldblum, M., Fowl, L., Terry, J.K., Huang, F. and Goldstein, T., 2020. Understanding generalization through visualizations.
>
> [2] Dinh, L., Pascanu, R., Bengio, S. and Bengio, Y., 2017, July. Sharp minima can generalize for deep nets. In the International Conference on Machine Learning (pp. 1019-1028). PMLR.
>
> [3] Petzka, H., Kamp, M., Adilova, L., Sminchisescu, C. and Boley, M., 2020. Relative Flatness and Generalization. arXiv preprint arXiv:2001.00939.
>
> [4] Liang, T., Poggio, T., Rakhlin, A. and Stokes, J., 2019, April. Fisher-rao metric, geometry, and complexity of neural networks. In The 22nd international conference on artificial intelligence and statistics (pp. 888-896). PMLR.

---

### Official Review · Reviewer_coDM · 2025-11-01

**Soundness:** 3
**Presentation:** 2
**Contribution:** 2
**Rating:** 2
**Confidence:** 4

**Summary:**

The core message here—flatness by itself is not a reliable predictor of generalization—isn’t new. The paper reframes this  as a “function‑centric view” and backs it up with a broad set of experiments (toy objectives, CIFAR‑10/100, Tiny‑ImageNet; ResNet/VGG/ViT) and a few safety‑relevant metrics (ECE, corruption accuracy, “prediction disagreement”). The coverage of datasets is solid, but the headline observation has been known since the reparameterization critique from Bengio's group  (and follow‑ups) made the “flat vs. sharp” narrative much more nuanced -- see e.g. an older paper from 2021 -Why flatness does and does not correlate with generalization for deep neural networks  https://arxiv.org/abs/2103.06219 -- for example

So what’s left is not that new.

Secondary mistakes:
They wrote “decisecond boundaries then base models” twice, It should me decision boundaries than base models. (Appendix E.2). Shows lack of proofreading.
The reference entry for Li et al., 2018 (Loss Landscape visualization) points to the Verified Uncertainty Calibration NeurIPS URL hash (Kumar et al., 2019). The link text and the target don’t match.
Booth function is misstated as (x+2y−y)^2+(2x+y−5)^2. It should be (x+2y−7)^2+(2x+y−5)^2

**Strengths:**

The coverage of datasets is solid,

The results are largely correct.

**Weaknesses:**

The paper repeats many things that are already pretty well known.  I suspect that they have not fully read or digested the literature on this topic.

**Questions:**

what is truly novel in this work?

---

> ### Author Response · Authors · 2025-11-18
> **Response 1**
>
> **Overall Comments**
>
> Thank you for taking the time to review our paper and digest its findings. We appreciate the acknowledgement of our solid coverage of datasets and architectures as well as the correctness of our findings. **We also appreciate the opportunity to further highlight the novelty of our work which we argue is a timely and interesting insight, which has also been acknowledged by reviewers 2Sqz, 2jxP and 29Eb.**
>
> **W.1 The paper repeats many things that are already pretty well known. I suspect that they have not fully read or digested the literature on this topic.**
>
> As stated in our introduction (L35 - L40), we address the tension within the literature of the importance or preference towards flat over sharp minima - an ongoing debate. Dihn et al’s [1]  paper, provided by Bengio’s group (2017), provides extremely strong evidence against classical Hessian-based sharpness/flatness metrics provided in the literature; this is not something we refute, but something that we acknowledge in our work. In fact, our work extends the analysis by Dihn et al and provides further insight on how function spaces can govern the properties of minima.
>
> Liang et al., in 2019 [3] and Pzeka et al., in 2021 [2] provided two metrics that were invariant to the layer-wise and neuron-wise reparameterisations provided by the original work produced by Dihn et al (2017). These metrics were specifically chosen for our study due to their ability to navigate the complexities introduced by reparameterisation, ensuring a more accurate assessment of flatness and its implications. The robustness of these metrics against reparameterisation was not only validated empirically, as shown in Figures 5 and 6 in the appendix of Pzeka et al., 2021 [2], but also presented a strong correlation between flatness and a reduced generalisation gap. This validation underpins our confidence in employing these metrics as a foundational element for our analysis.
>
> Furthermore, more recent literature by Stutz et al (2021) [4]  conducts a study which states that models that are adversarially robust models correlate with flatter minima, further reinforcing notions of general flat minima benefits. Cha et al (2021)  [5]  produce Stochastic Weight Averaging Densely (SWAD), which they argue improves domain generalisation by seeking out flat minima, which they use loss landscapes to visualise. Zhao et al (2022) [6]  build upon sharpness-aware minimisation to constrain the gradient norm to efficiently find flatter minima and reach models that generalise better.
>
> Most recently, in reinforcement learning, recent literature by Lee et al. (2025)  [6] argues that flatter minima improve the robustness of RL agents. Han et al. (2025) [7] posit that flatness is a necessary component of generalisation in the grokking setting, where there is a phase of extreme memorisation followed by generalisation. Finally, [8] (2025) also perpetuates the notion that flatness is a fundamental property of well-performing systems with small generalisation gaps.
>
> As a result, this work seeks to further explore the ongoing contention in the literature between flat and sharp minima. Given the  literature's strong bias towards flat minima, we leverage the very metrics that have been utilised to further validate the relationship between flatness and generalisation. In a controlled setting,  we test how regularisation methods, which are known to improve performance (weight decay, augmentation, and SAM), impact the geometry of a model's landscape compared to an appropriate baseline across architectures and datasets. We perform, to our knowledge,  one of the largest, most comprehensive empirical analyses of minima to provide insights into minima in practice under a range of different control conditions.
>
> What we observe is contrary to what would be expected based on current literature, since Dihn et al 2017 [1] . When performance increases under the application of these regularisers and the generalisation gap is reduced, we see increased sharpness over the baseline. Therefore, our results provide a necessary and timely contribution to the debate surrounding flat and sharp minima with implications for how we consider generalisation of deep neural networks.
>
> **Given the existing tension in the community on sharpness and flatness since Dinh et al, 2017, could you reconsider your position on our digestion of literature and the importance of this work?**
>
> **W.2 Secondary mistakes**
>
> Thank you for identifying the spelling mistake, we will update the paper to correct this. We also appreciate the identification of the incorrect term in the booth function definition, this will additionally be updated along with the link and target text in the bibliography. We will upload the final draft at the end of the rebuttal after responding to all reviewers requests for experiments.

---

> ### Author Response · Authors · 2025-11-18
> **Response 2**
>
> **Q1. what is truly novel in this work?**
>
> Thank you for the opportunity to highlight the novelty of our work.
>
> As far as we are aware, the core reason for generalisation in deep neural networks remains a mystery. As stated by reviewer 29Eb: “Since flatness remains one of the main candidates for explaining why deep networks generalize, this empirical study makes a meaningful contribution to an ongoing and relevant discussion in the field.”.
>
> While we do not claim to solve the mystery of why deep neural networks generalise, this work means that we can converge on what meaningful studies of generalisation to consider, as we show that generalisation is not causally related to flatness and that we need to consider both the complexity of the problem space and the type of optimisation processed used to begin to explain the properties of the types of minima that are converged to at the end of training.
>
> **Our experimental setup is one of the most extensive in the current literature exploring geometric properties of neural networks**. We explore four architectures (MLPs, ResNets, VGGs, and ViTs) across four task complexities (7 Single Objective Optimisation functions, CIFAR10, CIFAR100, and TinyImageNet). We use three independent regulariser controls: weight decay, augmentation, and sharpness aware minimisation. This approach provides deep insights into the nuance between performance increase, regularisation, and their interaction with task complexity.
>
> Additionally, we move beyond simple characterisations of flat minima and generalisation to characterise geometric properties with a range of safety-critical evaluations, including robustness, calibration, and prediction similarity, which are crucial for models in real-world deployment settings. We do not conduct our sharpness evaluations on single metrics but solely on the two core reparameterisation invariant metrics  provided by the literature to increase the robustness of our findings.
>
> Furthermore, analysing regularisation from a functional perspective provides new ways to understand how performance and minima sharpness increase under the application of regularisation techniques.  The results allow us to understand the role and interactions of regularisation, which can cause models to have more complex representations (precise under Occam’s razor [10] and a longer Minimum Description Length (MLD) [11]) to enable improved performance, as observed in our results.
>
> We reframe prevailing narratives about the benefits of flat minima [2,3,4,5,6,7,8,9]. This adds a further dimension to the considerations of neural network minima. Our perspective directly connects to fundamental theories such as Occam’s Razor [10] and the Minimum Descriptive Length [11], and considers function complexity as core to learning. Our work builds on the seminal work of Dihn et al. We show a disconnect between the causal role of flat minima and generalization, even for advanced sharpness metrics. The generality and depth of our findings ensure that the functional perspective of flat and sharp minima can advance debates on generalisation dynamics and the geometric properties of minima, making our work a timely and necessary contribution to the field.
>
> References:
>
> [1] Dinh, L., Pascanu, R., Bengio, S. and Bengio, Y., 2017, July. Sharp minima can generalize for deep nets. In the International Conference on Machine Learning (pp. 1019-1028). PMLR.
>
> [2] Liang, T., Poggio, T., Rakhlin, A. and Stokes, J., 2019, April. Fisher-rao metric, geometry, and complexity of neural networks. In The 22nd international conference on artificial intelligence and statistics (pp. 888-896). PMLR.
>
> [3] Petzka, H., Kamp, M., Adilova, L., Sminchisescu, C. and Boley, M., 2020. Relative Flatness and Generalization. arXiv preprint arXiv:2001.00939.
>
> [4] Stutz, D., Hein, M. and Schiele, B., 2021. Relating adversarially robust generalization to flat minima. In Proceedings of the IEEE/CVF international conference on computer vision (pp. 7807-7817).
>
> [5] Cha, J., Chun, S., Lee, K., Cho, H.C., Park, S., Lee, Y. and Park, S., 2021. Swad: Domain generalization by seeking flat minima. Advances in Neural Information Processing Systems, 34, pp.22405-22418.
>
> [6] Zhao, Y., Zhang, H. and Hu, X., 2022, June. Penalizing gradient norm for efficiently improving generalization in deep learning. In International conference on machine learning (pp. 26982-26992). PMLR.
>
> [7] Lee, H.K. and Yoon, S.W., 2025, April. Flat reward in policy parameter space implies robust reinforcement learning. In The Thirteenth International Conference on Learning Representations.
>
> [8]  Han, T., Adilova, L., Petzka, H., Kleesiek, J. and Kamp, M., 2025. Flatness is necessary, neural collapse is not: Rethinking generalization via grokking. arXiv preprint arXiv:2509.17738. (Accepted at NeurIPS 2025)

---

> ### Author Response · Authors · 2025-11-18
> **Response 3 (Final)**
>
> **More references:**
>
> [9] Shoham, N., Mor-Yosef, L. and Avron, H., 2025. Flatness After All?. arXiv preprint arXiv:2506.17809.
>
> [10] Hochreiter, S. and Schmidhuber, J., 1994. Simplifying neural nets by discovering flat minima. Advances in neural information processing systems, 7.
>
> [11] Hochreiter, S. and Schmidhuber, J., 1997. Flat minima. Neural computation, 9(1), pp.1-42.

---

### Author Response · Authors · 2025-11-27
**Reviewer Prompt**

We would like to thank the reviewers for the commitment that they have shown to the review process by evaluating our submission. The identified weakness and questions asked have provided a great opportunity to highlight the importance of our reframing of flat and sharp minima, as well as leading us to conduct more empirical experiments, which provided stronger evidence for our findings in the paper.

We'd like to highlight some particularly useful outcomes of this rebuttal process:

1) Resolving tensions with SAM optimisation process: Through a hyperparameter sweep of SAM's  ρ (rho) parameter, we were able to demonstrate that with small values of ρ, SAM can reduce flatness and improve in generalisation and safety evaluations against the baseline. However, these results do confirm that most often SAM (under the reparameterization invariant metrics used in this paper)  reaches sharper minima that are more performant across all baselines, and the sharpest model can be the best.

2) Highlighting the importance of using reparameterisation invariant sharpness metrics: Our use of Relative Flatness and Fisher Rao norm is a crucial element of our paper, as they show that even under metrics that have reaffirmed that flat minima generalise better (since the work of Dinh et al[1]) we find that sharp minima arise under the application of commonly used regularisation. A novel and timely contribution to the field.

3) Inspired by our work in Section 4, we show task complexity and function complexity links: We increase the number of disjoint samples in a class and record sharpness values, and show that as the number of disjoint training examples increases (under the same computational budget) that there is a linear increase in the sharpness of minima at the end of training. These results strongly affirm our function centric view of flat and sharp minima.

4) Highlighting the preservation of sharpness trends for augmented and training data: We show that the trend of statistically significant sharpness increases for models trained in the augmentation and augmentation + SAM controls continues to hold regardless of whether the augmented or standard dataset is used to calculate sharpness. Improving the validity of the findings we present in the paper.

We acknowledge that this period is particularly busy; however, we would like to request that you consider our in-depth responses such that we can ensure we have addressed all of your concerns and can further elaborate if required.

The feedback from your reviews will be integrated into the revised manuscript once we have confirmation on the above. We have chosen to do this to ensure our edits reflect the latest feedback and are in line with your requests.

Thank you again for your efforts in the review process.

References:

[1] Dinh, L., Pascanu, R., Bengio, S. and Bengio, Y., 2017, July. Sharp minima can generalize for deep nets. In International Conference on Machine Learning (pp. 1019-1028). PMLR.

---

### Author Response · Authors · 2025-12-02
**Revised Paper in Response to Reviewers Requests**

We appreciate the reviewers' comments and questions, which have improved our work.  We have now uploaded the revised PDF, which has integrated all suggestions provided by the reviews. In the PDF, edits are presented in Blue. While these edits do not change the findings of the original manuscript they are added to increase completeness and quell concerns of reviewers.

For ease of assessment, below we outline where the changes have been made in the PDF and which weaknesses/questions they resolve in order of appearance in the paper:

1. Reviewer 29Eb (W.4), we highlight on L33 that Pzeka et al is an empirical and theoretical body of work.

2. Reviewer 2jxP (W.3), we clarify on L46, L64, and L184-186 that our robustness study is on average-case corruptions over adversarial assessments.

3. Reviewer 2jxP (W.4.2) and 29Eb (W.3), we provide a statement on Figure 1, L70-73, cautioning on the potential of the Simpson’s paradox when interpreting trends observed across conditions.

4. Reviewer 2Sqz (Q.1), we merged the original contributions 1 and 2 (summarised findings) in L109-111 to improve the clarity of our contributions and reduce redundancy.

5. Reviewer 2Sqz (Q.2) and coDM (W.1) in Section 3 L155-L161, we explicitly state why we focus on using reparameterisation invariant metrics over non-invariant sharpness metrics and why this is important to study since the work of Dihn et al. 2017

6. Reviewers 2Sqz (W.1), 2jxP (W.1 & W.2) and 29Eb (W.1&W.6) we  discuss in L201-204 how toy settings have been leveraged to understand geometric properties of neural network, on L275-280 we further clarify that we do not use Relative flatness in Section 4 due to its requirement for locally constant labels and in L297-300 signpost the reader to the new Appendix Section I which shows that sharpness directly increases when task complexity is arbitrarily increased to show that insights from Section 4 directly relate to notions of function complexity and minima geometry.

7. Reviewers coDM (W.1), 2Sqz (W.2), and 2jxP (Q.2), we added a paragraph, L314-326, explicitly explaining function complexity, extending ideas of Occam’s Razor and Minimum Descriptive length to provide an explicit understanding of function complexity and the novelty of our work's position and analysis.

Furthermore, we included the result we present in the rebuttal to 2jxP (Q.2), Appendix Section I, which shows explicitly an increase in function complexity relating to sharper minima found at the end of training.

8. Reviewer coDM (W.1) on L328-330, we add further justification for our work exploring different perspectives on minima since the work of Dhin et al.

9. Reviewer 2Sqz (Q3), we signpost the reader in the main body on L380-382 to our new appendix section on varying the $\rho$ hyperparameter of SAM, which provides more nuance on the geometric relationship between SAM and the perturbation radius. These results help to better explain our increased sharpness observed when using SAM.

10. Reviewer 2jxP(Q3), all loss landscape visualisation captions have been updated to better explain the exploration of the loss in the domain of perturbations $[-1,1]^2$ with 51 steps in both directions. See Tables 4, 6, 10, 19, 22 and 24.

11. Reviewer coDM (W.2), we updated the definition of the booth function in Appendix A L701 to remove the typo of -y and replace it with -7, updated the target link for Li et al., 2018 (L616)  to be correct, and fixed the decision boundary typo in the Appendix on L1055 and 1109.

12. Reviewer 2Sqz (Q4) in Appendix Sections B and C, we have added all formal definitions for completeness to complement our existing descriptions and target code links.

13. Reviewer 's 2jxP (W.4.1) and 29Eb (W.2) we added the sharpness calculations on the VGG architecture for CIFAR10 in Appendix F.2, which complement our preexisting findings with the loss landscape visualisations and results for the ResNet in Appendix Section E.1, that regardless of calculating sharpness on the standard training data or augmented data that models trained with Augmentation and Augmentation + SAM remain substantially sharper than any other condition, especially compared to the baseline.

**Overall**

Together, these revisions not only enhance the coherence of our work but also contribute meaningfully to ongoing efforts to unify geometric, optimization-based, and functional theories of neural network behavior. This is core to the field because it links the geometry of the loss landscape to the expressive capacity of the learned function, providing a novel explanation for why minima with similar sharpness or flatness can yield markedly different generalization outcomes. By grounding geometric interpretations in function-level behaviour, our work helps reconcile discrepancies across sharp and flat minima and offers a more stable foundation for understanding generalisation in modern deep networks beyond flat-minima–centric perspectives.

---

### Meta-Review · Area_Chair_SpXS · 2026-01-09

**Summary:**

This paper investigates the relationship between loss landscape geometry and model performance, arguing through reparameterization-invariant metrics that sharp minima can correlate with better generalization and safety, particularly under regularization. While the authors present an extensive suite of experiments, the paper is recommended for Rejection. The primary grounds for this decision are significant concerns remaining regarding the interpretation of "function complexity"; the authors' claim that regularization leads to "more complex" functions (sharper minima) contradicts standard learning theory where regularization is understood to restrict the hypothesis class (simplicity). Moreover, comparing sharpness metrics across fundamentally different loss landscapes (e.g., augmented vs. non-augmented objectives) presents methodological difficulties in isolating geometry from the objective function itself, which the rebuttal did not fully resolve to the reviewers' satisfaction. Despite the current recommendation, this work explores a valuable direction and presents promising preliminary results. I hope the authors are not discouraged by this outcome, but rather leverage the reviewers' feedback to strengthen the manuscript for a future resubmission.

**Reviewer Concerns:**

*** ADDRESSED

Technical Validation of SAM Trends (Reviewers 2Sqz, 29Eb): Reviewers noted that SAM is designed to minimize sharpness, contradicting the paper's findings. The authors successfully clarified this by performing a hyperparameter sweep, showing that while SAM can flatten minima at low radii, the specific high-performing configurations found in their study do yield higher sharpness values on the metrics used .

Calculation Consistency (Reviewers 2jxP, 29Eb): Reviewers questioned if sharpness metrics were calculated consistently on augmented vs. standard data. The authors provided appendix results showing that the trends (regularization increasing sharpness) remain statistically significant regardless of the data subset used for calculation .

*** OUTSTANDING

Novelty (Reviewer coDM): The reviewer argued that the paper "repeats many things that are already pretty well known" regarding the nuances of flatness. The authors' defense (that they are debunking the specific invariant metrics that emerged after Dinh et al.) was not seen as a sufficient leap.

Conceptual Definition of Complexity (Reviewer 2jxP): The fundamental tension between the authors' definition of complexity (sharper = more complex/precise) and the standard view of regularization (regularization = simpler functions) remains a barrier. The reliance on Occam's Razor/MDL to justify "sharpness as complexity"  was viewed as a semantic redefinition rather than a resolution of the contradiction with classical learning theory.

Methodological Validity of Cross-Landscape Comparisons (Reviewer 29Eb): Reviewers noted that changing the training objective (e.g., via augmentation or weight decay) alters the loss landscape itself. Therefore, comparing sharpness values between these conditions (e.g., Baseline vs. Augmentation) is arguably an invalid "apples-to-oranges" comparison, regardless of the metric used.

**Reviewer Scores:**

Reviewer 2Sqz (2): This reviewer would likely maintain their low score. Their fundamental questioning of the motivation behind Section 4 (comparing different objectives) and the contradiction with established SAM literature suggests the rebuttal did not alter their view on the paper's soundness.

Reviewer coDM (2): This reviewer would maintain their rejection. Their criticism was based on the lack of novelty relative to the 2017-2021 literature; the authors' rebuttal defended the work's position but did not demonstrate a fundamentally new theoretical breakthrough that would overturn the "incremental" assessment.

Reviewer 29Eb (4): This reviewer would likely remain at a reject/borderline. While their point about concurrent work was procedurally incorrect (as noted by the authors), their substantive technical point regarding "Simpson's Paradox" and the validity of comparing metrics across changed landscapes remains a valid reason for rejection.

Reviewer 2jxP (6): This reviewer might maintain a weak accept or drop to a borderline reject upon seeing the consensus. While they appreciated the paper's perspective, they also flagged the "Simpson's paradox" risks and the conceptual contradictions in regularization, which align with the reasons for the final rejection decision.

---

### Decision · Program_Chairs · 2026-01-26

Reject